# Targeting Toll-like receptor-driven systemic inflammation by engineering an innate structural fold into drugs

Ganna Petruk [1,10] ✉, Manoj Puthia [1,10], Firdaus Samsudin[2], Jitka Petrlova [1], Franziska Olm [3], Margareta Mittendorfer[3], Snejana Hyllén [3,4], Dag Edström[3,4], Ann-Charlotte Strömdahl[1], Carl Diehl[5], Simon Ekström [6], Björn Walse [5], Sven Kjellström [7], Peter J. Bond [2,8], Sandra Lindstedt[3,4] & Artur Schmidtchen [1,9]

There is a clinical need for conceptually new treatments that target the excessive activation of inflammatory pathways during systemic infection. Thrombin-derived C-terminal peptides (TCPs) are endogenous anti-infective immunomodulators interfering with CD14-mediated TLR-dependent immune responses. Here we describe the development of a peptide-based compound for systemic use, sHVF18, expressing the evolutionarily conserved innate structural fold of natural TCPs. Using a combination of structure- and in silico-based design, nuclear magnetic resonance spectroscopy, biophysics, mass spectrometry, cellular, and in vivo studies, we here elucidate the structure, CD14 interactions, protease stability, transcriptome profiling, and therapeutic efficacy of sHVF18. The designed peptide displays a conformationally stabilized, protease resistant active innate fold and targets the LPS-binding groove of CD14. In vivo, it shows therapeutic efficacy in experimental models of endotoxin shock in mice and pigs and increases survival in mouse models of systemic polymicrobial infection. The results provide a drug class based on Nature´s own anti-infective principles.

Systemic bacterial infections are characterized by activation of the innate immune system, initiated by simultaneous recognition of multiple pathogen-associated molecular patterns (PAMPs) and endogenous danger signals by various pattern recognition receptors, such as CD14 and Toll-like receptors (TLRs)[1,2]. TLR signaling and activation of nuclear factor kappa B (NF-κB) trigger a cascade of events leading to a systemic inflammatory response syndrome, involving a massive release of cytokines, acute phase proteins, and reactive oxygen species[1,3–6].

This dysregulated inflammation causes organ dysfunction, such as acute respiratory distress syndrome (ARDS), a characteristic feature of sepsis[7]. Sepsis contributes significantly to mortality in intensive care units, resulting in substantial healthcare costs[8]. Globally, it affects a staggering number of individuals, with approximately 50 million cases reported each year. Age-standardized mortality rates reveal that sepsis accounts for nearly 20% of all deaths, resulting in approximately 11 million fatalities annually[9,10].

[1]Division of Dermatology and Venereology, Department of Clinical Sciences, Lund University, SE-22184 Lund, Sweden. [2]Bioinformatics Institute (BII), Agency for Science, Technology and Research (A*STAR), Singapore 138671, Singapore. [3]Department of Clinical Sciences, Lund University, SE-22184 Lund, Sweden. [4]Department of Cardiothoracic Surgery, Anesthesia and Intensive Care, Skåne University Hospital, SE-22185 Lund, Sweden. [5]SARomics Biostructures AB, Medicon Village, SE-22381 Lund, Sweden. [6]BioMS - Swedish National Infrastructure for Biological Mass Spectrometry, SE-22184 Lund, Sweden. [7]Division of Mass Spectrometry, Department of Clinical Sciences, Lund University, SE-22184 Lund, Sweden. [8]Department of Biological Sciences, National University of Singapore, Singapore 117543, Singapore. [9]Dermatology, Skane University Hospital, SE-22185 Lund, Sweden. [10]These authors contributed equally: Ganna Petruk, Manoj Puthia. ✉e-mail: ganna.petruk@med.lu.se

Efforts to treat sepsis solely by eliminating the invading pathogen have proven insufficient, as sepsis mortality remains high despite antibiotic treatment. It is crucial to note that antibiotics do not specifically target the excessive and harmful activation of inflammatory cascades triggered by bacteria and bacteria-derived PAMPs such as lipopolysaccharide (LPS).

Consequently, multiple randomized controlled clinical trials employing other therapeutic compounds have been conducted in sepsis patients. However, apart from activated protein C, these all failed to significantly improve survival. Subsequently, activated protein C was withdrawn from the market due to lack of efficacy[11–13]. Importantly, systemic inflammation, which underlies sepsis symptoms, remains inadequately addressed, necessitating the development of alternative therapeutic approaches to improve clinical outcomes.

Injury and infection are serious threats to the survival of an organism, and nature has therefore evolved powerful natural peptide-based mechanisms for controlling excessive bacterially-induced inflammation. Peptides derived from the C-terminus of thrombin (TCPs), initially discovered in human wounds[14], control both bacterial infections and the associated inflammatory responses[15–17] via transient and multiple interactions with Gram-positive and negative- bacteria and their PAMPs, and subsequent interference with CD14-mediated TLR signaling[15–17], neutrophil chemotaxis[18], and excessive contact activation[17].

Supported by preclinical data demonstrating that the TCP GKY25 (with sequence GKYGFYTHVFRLKKWIQKVIDQFGE) effectively treats infections and excessive inflammation in murine and porcine wounds, improving healing[19,20], GKY25 is currently undergoing clinical development as a topical treatment for various wound indications[21]. The observation that GKY25 also reduces inflammation and mortality in animal models of systemic endotoxin shock and bacterial infection[14,17] indicates that the drug concept could have clinical applications far beyond wound healing. However, like many peptide-based therapeutics, GKY25 is susceptible to proteases[19,20]. While generation of an endogenous "peptidome" is advantageous in wounds[19,22], yielding an overall boosting of natural host defense mechanisms[19], such degradation is less preferable after systemic administration as it would require prohibitively large doses and frequent administration, and complicate the clinical and regulatory development work.

With this background, we therefore set out to use structural, functional, and evolutionary information for the development of a peptide-based drug class aimed for systemic use, expressing the evolutionarily conserved innate structural fold of natural TCPs in their active stabilized form. Employing a combination of structure and in silico-based design strategies, nuclear magnetic resonance (NMR) spectroscopy, hydrogen-deuterium exchange mass spectrometry (HDX-MS), molecular modeling, and biophysical, cellular, and in vivo studies, we describe the successful development of a peptide-based compound, here denoted sHVF18, that specifically targets the LPS-binding grove of CD14, enabling inflammation modulation. sHVF18 is protease-resistant, ameliorates inflammation and lung damage in mouse and pig models of endotoxin shock, and reduces mortality in mouse models of systemic polymicrobial infection.

## Results
### Design strategy for TCP peptide mimics
Stapling can significantly increase proteolytic resistance, target affinity, cell permeability, and plasma half-lives in vivo[23–25]. One recent approach involves peptide hydrocarbon stapling, which can stabilize secondary structure with a side-chain covalent hydrocarbon bridge[26]. The TCP-sequence KKWIQK binds to the LPS-binding hydrophobic pocket sequence of CD14, with the TCP-region overlapping with amino acid residues involved in LPS interaction (RLKKWIQK)[15]. We decided to adopt a strategy enabling the preservation and boosting of TCP's core action towards a dual LPS and CD14 interaction, enabling multimodal

control of inflammation[14–17,19,27]. Based on structural and functional clues, together with in silico analyses of stapling positions and CD14 binding energies, stapling of Q17 and D21 was found to be optimal to generate a site mimicking the active conformation of LPS-bound and CD14 interacting TCPs, such as GKY25 and HVF18[15] (Fig. 1a, Supplementary Note 1, Supplementary Fig. 1). CD analysis of stapled GKY25 (sGKY25) indicated a structural conformation similar to the linear counterpart in complex with LPS (Supplementary Fig. 2a). sGKY25-showed an improved protease resistance at its C-terminal part, comprising the HVF18 sequence (Supplementary Fig. 2b, c, Supplementary Table 1). Moreover, the peptide exhibited a significantly improved inhibitory activity on LPS-stimulated THP1-XBlue-CD14 reporter cells (Supplementary Fig. 2d). However, it also displayed significant hemolysis and, in connection to this, also a reduced anti-inflammatory effect in LPS-stimulated whole blood (Supplementary Fig. 2e, f), which precluded further development. Next, a two-pronged approach was adopted to generate a new generation of stapled TCPs (Fig. 1b). First, as a covalent link at the N-terminus in position 1 and 5 was compatible with CD14 binding (Supplementary Fig. 1b) and could protect from aminopeptidases, this feature was added to sGKY25. Second, as proteolysis of sGKY25 indeed generated a sHVF18 sequence, we concluded that this protected part could constitute an attractive mimic, as it contained the critical features governing peptide efficacy[15]. In addition, we also synthesized N-terminally truncated variants, including the minimal CD14 interacting region KKWIQK (sKKW13). We analyzed the hemolytic effects in human blood and on red blood cells (RBC) and anti-inflammatory activity in LPS-stimulated human blood (Fig. 1c, d). The correlation between this peptide's respective $IC_{50}$ for cytokines (TNF-α and IL-1β) in LPS-stimulated blood and its hemolytic activity on isolated RBCs is shown in Fig. 1e. Taken together, sHVF18 demonstrated the lowest $IC_{50}$ values of the variants (around 5 μM), and the low RBC hemolysis was reflected by the low hemolysis in blood by the peptide at 50–100 μM. Hence, by defining the therapeutic index in human blood as the ratio between toxicity and efficacy, sHVF18 showed an over 10-fold improvement.

### Biophysical and biochemical characterization of sHVF18
Circular dichroism (CD) yields information on peptide secondary structure. We found that sHVF18, in contrast to linear HVF18, yielded a significant α-helical signal compatible with its predicted conformation at the peptide's C-terminal stapled end. Both the linear and stapled peptides showed similar α-helical signals in the presence of LPS (Fig. 2a). It is well established that the α-helical secondary structure affects peptide hydrophobicity, and thus interactions with reversed phase matrices[28–31]. Correspondingly, analysis of linear and stapled HVF18 using reversed-phase high-performance liquid chromatography (RP-HPLC) showed a longer retention time for sHVF18 (9.42 min) when compared with linear HVF18 (8.03 min), confirming that structural "locking" increased the hydrophobic binding region of sHVF18 (Supplementary Fig. 3a).

We next determined the effects of various proteases on the two peptides using a combination of SDS-PAGE (Fig. 2b), RP-HPLC (Supplementary Fig. 3b) and LC-MS/MS (Fig. 2c and Supplementary Table 2). The results showed that sHVF18 had improved resistance against the actions of human neutrophil elastase (HNE), *P. aeruginosa* elastase (PE), as well as trypsin (Try). Analysis by LC-MS/MS of HNE and PE-treated peptides demonstrated a significant difference in fragmentation, with no identified sHVF18 fragments after 30 min and 3 h of digestion (Fig. 2c and Supplementary Table 2). The conformational stability of sHVF18 was further analyzed by exposure of the peptide to 80 °C followed by CD analysis (Supplementary Fig. 4). We found that sHVF18 still presented an α-helical spectrum with two characteristic minima at 208 and 222 nm. The conformational flexibility of HVF18 after stapling was evaluated using HDX-MS[32]. The results reported in Fig. 2d (left panels) show that HVF18 becomes fully deuterated already

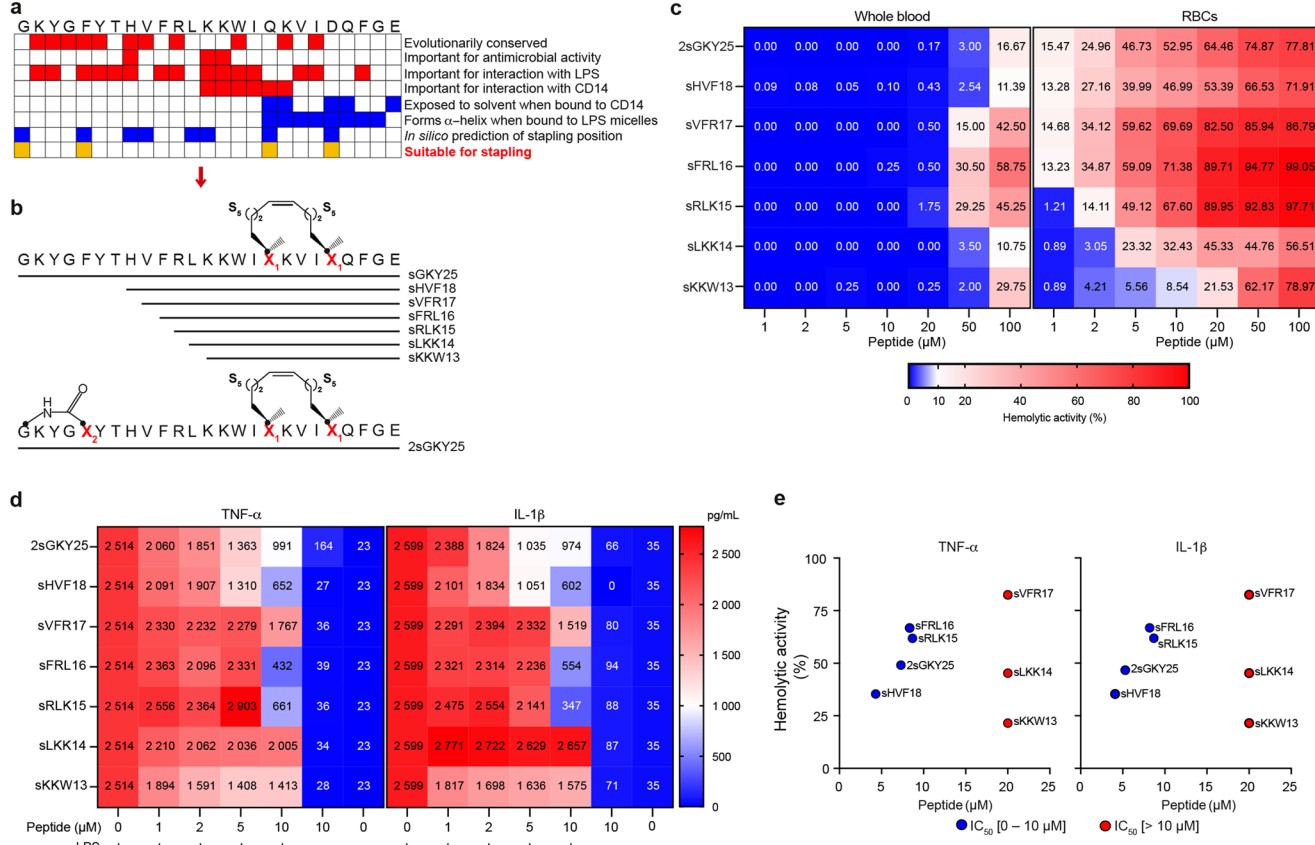

**Fig. 1 | Selection of the stapled peptide with improved anti-inflammatory activity. a** Selection criteria for stapling positions. **b** Illustration of peptide sequences. **c** The heatmaps show the hemolytic activity of the peptides on whole blood or erythrocytes (RBCs). Data from experiments performed on erythrocytes or blood from different donors are presented as mean ($n \geq 4$). **d** The heatmaps show the cytokines released from human blood stimulated with 100 ng ml$^{-1}$ *E. coli* LPS in the presence or absence of increasing concentrations of different peptides 24 h post-stimulation. Results are presented as mean values. Blood from different donors was used each time ($n \geq 4$). **e** Graphs obtained combining data from (**c**) and (**d**) represent the hemolytic activity of the peptides as a function of their IC$_{50}$ for different cytokines. Hemolytic activity at 20 μM is shown for peptides with IC$_{50}$ > 10 μM.

at the 30-s label time, as indicated by an observed uptake of 9 deuterons (D) with no further increase at longer labeling times, which is typical for high flexible or disordered species, with little or no secondary structure involving hydrogen bonds present to protect from the exchange. Conversely, for the stapled peptide, the deuterium uptake increases with time (from 2D at 30-s to 6D at 9000-s), which most likely can be attributed to increased conformational stability (Fig. 2d, right panels, Supplementary Data 1).

**NMR analysis of sHVF18 structure**

To gain further insight into the stapled peptide conformation, we performed a detailed nuclear magnetic resonance (NMR) study of sHVF18. Initial experiments, as well as TOCSY, NOESY, ROESY, $^{13}$C-HSQC, and $^{15}$N-SOFAST-HMQC spectra for peptide in 50% TFE, are described in Supplementary Note 2. The secondary structure of sHVF18 was estimated (Fig. 2e). The 3D structural ensemble of sHVF18 was calculated using distance restraints derived from NOE cross peaks and dihedral angle restraints derived from backbone chemical shifts. The peptide forms an L-shaped structure with two α-helices, consisting of residues V2-L5 and K11-G17. The N-terminal and C-terminal α-helices are structurally well-defined, but there is significant variability in the orientation of the two α-helices with respect to one another (Fig. 2f). When the structure of sHVF18 in TFE is compared to HVF18 in the presence of LPS, it is possible to observe a similar L-shaped tertiary structure, with a pairwise backbone root mean square deviation (RMSD) of 2.2 Å (Fig. 2g). The main differences are seen for the N-terminal part of HVF18, where the α-helix seen in sHVF18 is not

observed. This may, however, be due to TFE inducing sHVF18 into a more helical structure than LPS induces in HVF18, as shown by CD (Supplementary Fig. 9). In sHVF18, the N-terminal part of sHVF18 seems to be more ordered than in HVF18. The C-terminal α-helix in HVF18 ranges over residues I9-G17, which is two amino acids longer than sHVF18. A more detailed look at the NOEs acquired for sHVF18 and a comparison of these to the distances in the HVF18 structure ensemble reveals some differences for residues L5, K7, W8, I9, and K11. This was not unexpected, given the different NMR conditions and structural constraints imposed by stapling.

**Effects of peptide stapling on CD14 and LPS interactions**

We previously showed that HVF18 has a higher affinity for CD14 than observed for GKY25, whereas the reciprocal applies to the two peptides' interactions with LPS[15,17]. The restriction of HVF18 motion upon binding to CD14 is expected to cause a loss of configurational entropy and, thus, a penalty in binding affinity[33].

Based on this assumption, designing a conformationally locked peptide should, therefore, lead to an affinity gain for CD14. To study this, we exploited microscale thermophoresis (MST), a highly sensitive technique for probing interactions between components in solution. As predicted, the $K_d$ of sHVF18 to CD14 was significantly reduced when compared with linear HVF18 and GKY25, even in the presence of salt (Fig. 3a). The higher affinity of sHVF18 to CD14 is reflected by the peptide's significantly increased inhibition of NF-κB activation in LPS-stimulated THP1-XBlue-CD14 reporter cells (Fig. 3b). A similar activity increase was observed for the TLR-agonists LTA and PGN from

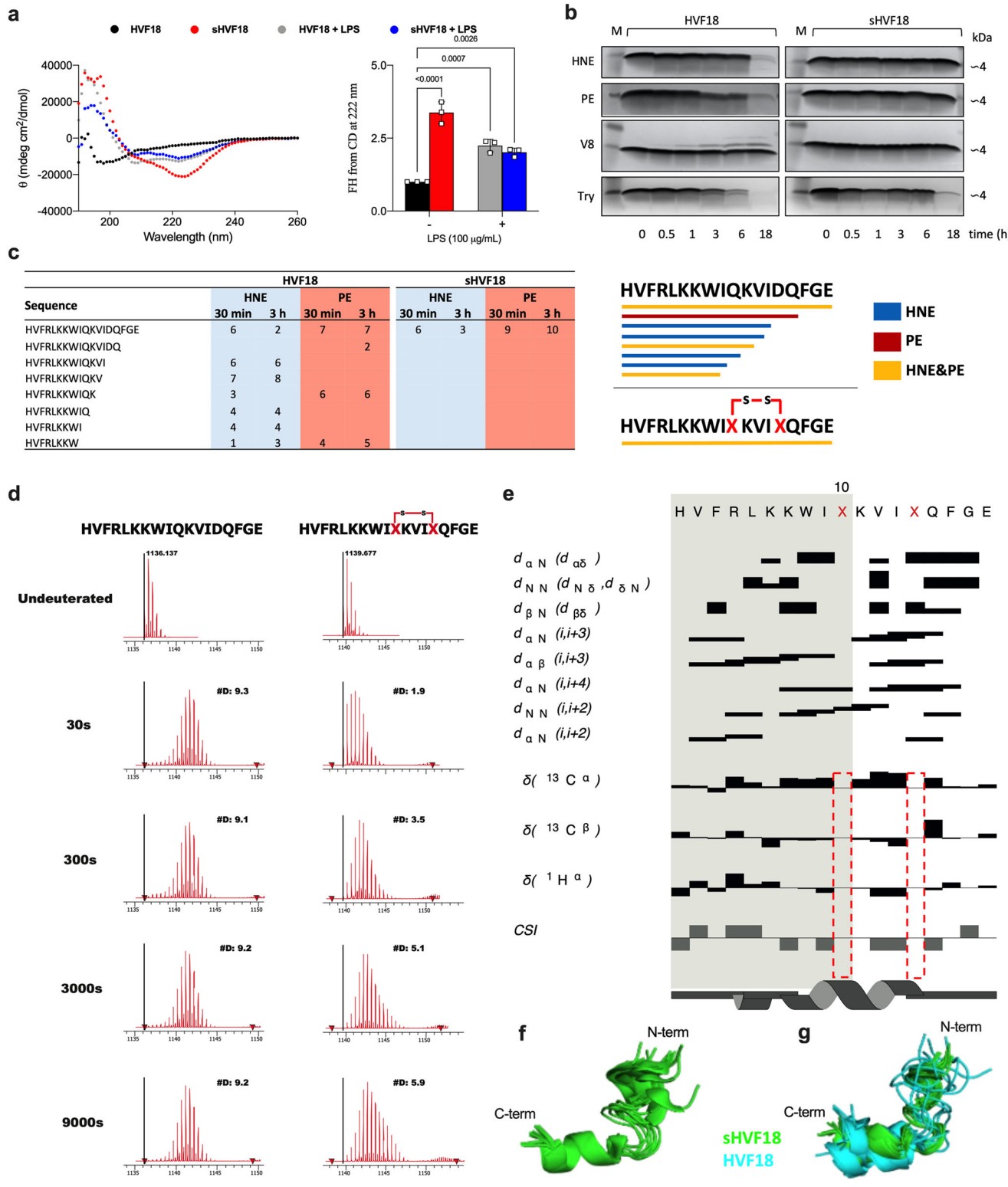

**Fig. 2 | Biophysical and biochemical analysis of stapled HVF18. a** Representative CD spectra of linear and stapled HVF18 in the absence and presence of LPS. The relative α-helical content was calculated from CD spectra. Each data point was obtained from a separate experiment, data are presented as mean ± SEM ($n = 3$). *P*-values were determined by an ordinary two-way ANOVA followed by Tukey's multiple comparisons tests using GraphPad Prism software. **b** SDS-PAGE of intact peptides and peptides digested with the indicated proteases for different lengths of time. One representative image from three independent experiments is shown ($n = 3$). **c** Table and graphical representation summarize all the major peptides detected by LC-MS/MS after digestion with human neutrophil elastase (HNE) and

*Pseudomonas* elastase (PE) for 30 min and 3 h. Experiments were performed two times in duplicate, using two different digestions ($n = 2$). **d** Deuterium uptake by HVF18 (left) and sHVF18 (right) upon different exposure times. Experiments were performed twice with similar results ($n = 2$). **e** Secondary structure chart of sHVF18 depicts inter-residue NOE cross peaks, chemical shift differences from the random coil and chemical shift index (CSI). The depiction of the secondary structure comes from DANGLE dihedral angle estimations. The structure was calculated on peptide dissolved in 50 % TFE. **f** Ensemble structure in cartoon representation of the backbone of sHVF18 based on NMR data. **g** Overlaid structure ensembles of the sHVF18 (green) and HVF18 (blue) peptides.

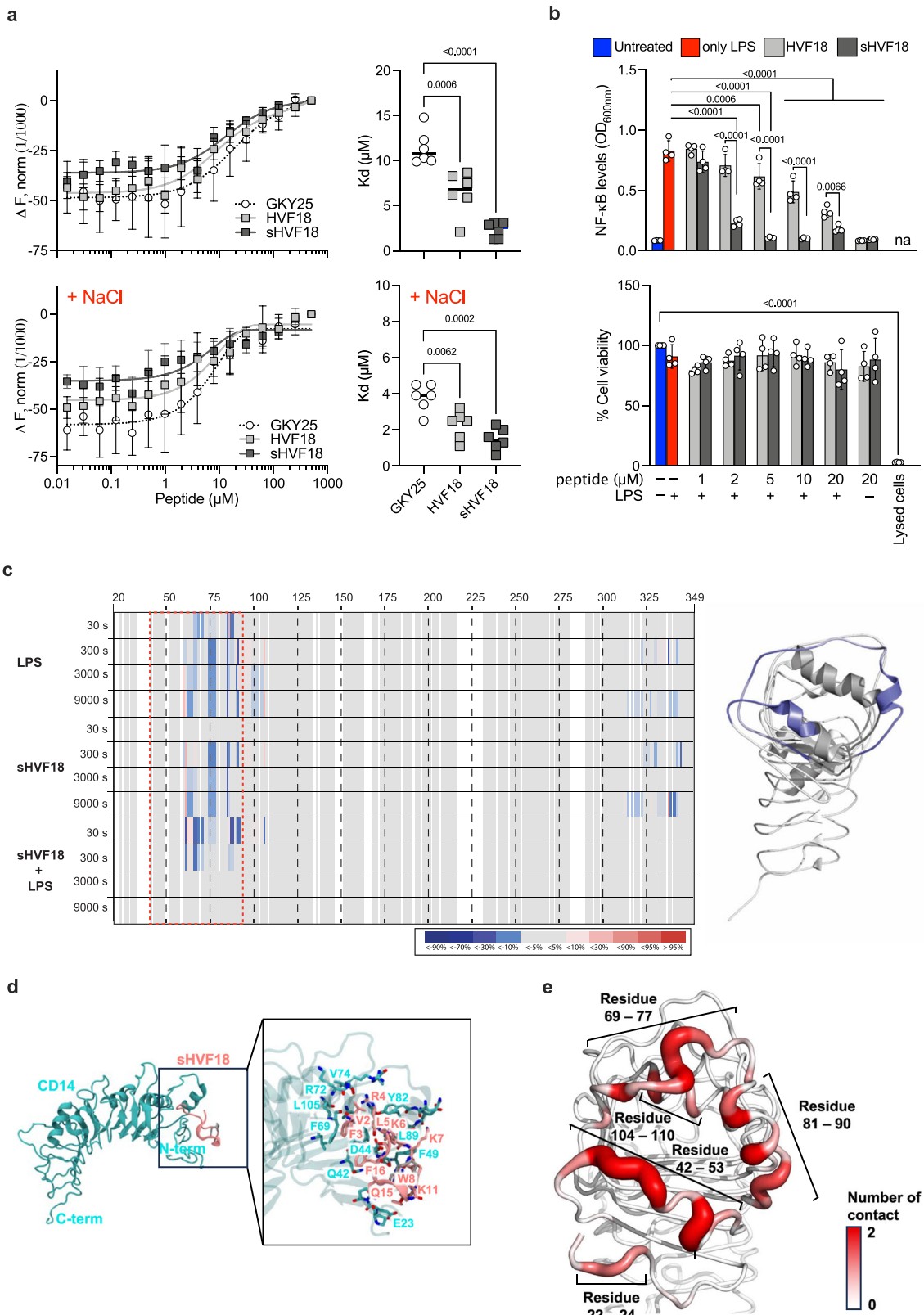

S. aureus, PGN from E. coli and zymosan from S. cerevisiae (Supplementary Fig. 10).

Next, to map the interaction sites for LPS and sHVF18 on CD14, we employed HDX-MS. Briefly, measuring the exchange rate of amide bond hydrogens as the increase of mass upon exposure to deuterated buffer by mass spectrometry can provide information regarding structure, protein-protein interactions, allosteric effects, intrinsic disorder, and conformational change. In this case, if a ligand interacts with CD14, the potential interaction sites can be identified based on their lower uptake of deuterium (protection) than observed for the protein alone[34]. The X-ray crystal structure for soluble human CD14 has previously been solved, and a binding site for LPS proposed, with the main interaction traced to four functionally important regions delineated by amino acid residues 26–32, 41–44, 56–64, and 78–83[35]. The

**Fig. 3 | Effects of stapling on CD14 binding. a** Representative binding curves between CD14 and GKY25, HVF18 or sHVF18, in the presence or the absence of NaCl, obtained by MST. $K_d$ values were calculated from MST curves. The data are presented as mean ± SD of six different measurements (*n* = 6). *P*-values were determined using an ordinary one-way ANOVA followed by Tukey's multiple comparisons tests using GraphPad Prism software. **b** NF-κB activation and cell viability in THP-1 monocytes stimulated with 100 ng ml$^{-1}$ *E. coli* LPS in the presence or absence of increasing concentrations of HVF18 and sHVF18, 20 h post-stimulation. Results are presented as means ± SD of four experiments (*n* = 4). *P*-values were determined using an ordinary two-way ANOVA followed by Tukey's multiple comparisons tests using GraphPad Prism software. **c** Differential heatmaps

for CD14 − LPS, CD14 − sHVF18, and CD14 − LPS + sHVF18. Heatmaps show the differential deuterium uptake at the different labeling times (3, 300, 3000, and 9000 s) based on volcano plot significance lines (95% CI) for comparing the respective states (see Supplementary Data 1). (Right) Commonality in protection denoted by red box superimposed on PDB:4GLP [https://www.rcsb.org/structure/4GLP] (note the structure starts at residue 26). **d** The central structure of the top cluster from a clustering analysis performed on concatenated trajectories of three independent 1 μs simulations. CD14 is shown in cyan and the peptide is in pink with the staple in grey. The enlarged image highlights key residues for peptide binding from simulations. **e** Average number of contacts made by residues on CD14 with sHVF18 during simulations. The cut-off distance for contact analysis is 0.4 nm.

HDX data presented here support these LPS binding sites, and protection was mainly observed in the N-terminal rim residues, spanning F38-L106, followed by protection at longer label times for the region L123-W160 and downwards, indicative of a conformational change in CD14 upon LPS binding. (Supplementary Fig. 11). Interestingly, the HDX data also showed that sHVF18 (and HVF18) appears to interact mainly with the N-terminal region of CD14 (blue areas coved by red box in Fig. 3c, and Supplementary Fig. 12). When LPS and sHVF18 were added simultaneously to CD14, the protection at the delineated LPS binding site decreased, and was significant only at 30–300 s (Fig. 3c, and Supplementary Fig. 13). While the similarity in N-terminal engagement between LPS and peptide is indicative of competitive binding, the decreased protection observed when both are added simultaneously could be due to LPS-sHVF18 interactions.

To further investigate the mode of interaction of sHVF18 with CD14 as well as LPS, we first addressed the CD14 binding and performed unbiased in silico docking and molecular dynamics (MD) simulations. The structure of the sHVF18 peptide solved using NMR spectroscopy described above was first docked onto the structure of human CD14. To sample as many potential docking poses as possible, no intermolecular restraints were applied during the peptide-protein docking protocol. The majority (11 out of 17) of docking poses were located at the N-terminal hydrophobic pocket (Supplementary Fig. 14a), previously proposed to be the binding site for LPS and linear TCPs[15]. We first performed 200 ns simulations for each of the 11 docking poses around the N-terminal pocket, and encouragingly the peptide remained bound to the protein in all simulations (Supplementary Fig. 14b). The most representative structure across all simulated poses was then determined using clustering analysis, and was found to comprise ~90% of all frames. This representative structure was then subject to extended 1 μs simulations (details in Methods). The sHVF18 peptide remained bound throughout three independent repeats simulations, with the N-terminal residues consistently anchored to the binding site (Supplementary Fig. 14c). In contrast, the flexible C-terminus was more mobile, as indicated by higher RMSD values. Measurement of the per-residue solvent accessible surface area (SASA) for the peptide revealed the highest SASA values for the C-terminal E18 residue (Supplementary Fig. 14d), in agreement with the increased flexibility of the C-terminus. Further clustering analysis of these longer trajectories highlight extensive interactions with residues on the N-terminal rim of CD14 that facilitate sHVF18 binding during the simulations including D44, F49, V74, Y82, L89 and L105 (Fig. 3d). The key residues involved in binding are similar to those previously proposed for linear TCPs and for LPS[8]. The residues on sHVF18 that are involved in binding include K6, K7, W8 and K11, which form the evolutionarily conserved TCP motif KKWIQK that has previously been shown to cross-link with CD14[15]. Contact analysis revealed stretches of residues on the N-terminal rim of CD14 that made significant interactions with sHVF18, primarily residues 42-53, 69-77, 81-90 and 104-110 (Fig. 3e). Our independently performed docking and simulations are consistent with the above HDX-MS data showing deuterium exchange protection for CD14 peptides spanning residues F38-L106 (Fig. 3c and Supplementary Fig. 12a, b). We note that our HDX-MS results also

suggest some allosteric effects of sHVF18 binding on CD14, although longer simulations would be required to observe these effects in silico (Supplementary Note 3). We acknowledge that while our HDX and in silico data provide a guide to the potential binding site of sHVF18 on CD14, more detailed structural studies may resolve a higher-resolution binding pose of the peptide.

As mentioned above, our HDX data also demonstrate weakened protection of the putative binding site in the presence of both LPS and sHVF18, suggesting LPS-sHVF18 interactions. We thus performed coarse-grained (CG) MD simulations of sHVF18 in the presence of lipid A (the primary bioactive lipid-containing component of LPS) (Supplementary Fig. 15, Supplementary Note 4). We found that similar to the linear peptide[8], sHVF18 spontaneously adsorbed onto the surface of a lipid A aggregate. We also repeated these CG MD simulations with an *E. coli* rough Ra LPS lipid aggregate, and observed similar binding of sHVF18 (Supplementary Fig. 16, Supplementary Note 4). In agreement, a microscale thermophoresis (MST) binding assay revealed that sHVF18 binds to LPS with a Kd of 2.5 ± 0.7 μM, while its linear variant binds with a Kd of 4.7 ± 0.8 μM (Supplementary Fig. 17). Thus, the observed LPS-sHVF18 binding is consistent with the weakened protection of the binding site on CD14 in presence of LPS (Fig. 3).

Our results were further verified by atomistic MD simulations showing similar interactions of sHVF18 to a lipid A micelle (Supplementary Fig. 18, Supplementary Note 4). We performed structural analysis to compare the structures derived from our MD simulations of lipid A-bound sHVF18 to the NMR structure of the peptide in 50% TFE. Clustering analysis highlights similar structural features, for example, two short stretches of alpha helices on the N- and C-termini of the peptide, and a more dynamic N-terminus (Supplementary Fig. 18a). Additionally, the backbone RMSD of the peptide after least square fitting to the NMR ensemble remained low across our simulation trajectories at around 0.5 nm (Supplementary Fig. 18b), while the peptide retained most of its helicity throughout (Supplementary Fig. 18c). This structural analysis thus suggests that the TFE-derived NMR structure is similar to the physiologically relevant LPS-bound state of the peptide. Taken together, our in silico docking and simulations provide further support that sHVF18 binds to CD14 at the same site as LPS, thus interfering with LPS binding and onward transfer, while simultaneously being able to neutralize LPS aggregates in solution. It is noteworthy that residues in the evolutionarily conserved TCP innate sequence KKWIQK interact with both CD14 and LPS in our simulations, emphasizing the significance of structurally stabilizing this particular region by addition of the peptide staple.

## Effects of sHVF18 on LPS responses in experimental mouse model

We next wanted to evaluate the efficacy of sHVF18 in two endotoxin-induced inflammation models. In the first model, the purpose was to explore whether sHVF18 could suppress LPS-induced local subcutaneous inflammation in vivo as well as to compare the effects relative to the linear peptide side-by-side. Using mice reporting NF-κB activation, we found that subcutaneous injection of LPS yielded a local inflammatory response, which was abrogated by sHVF18 during the

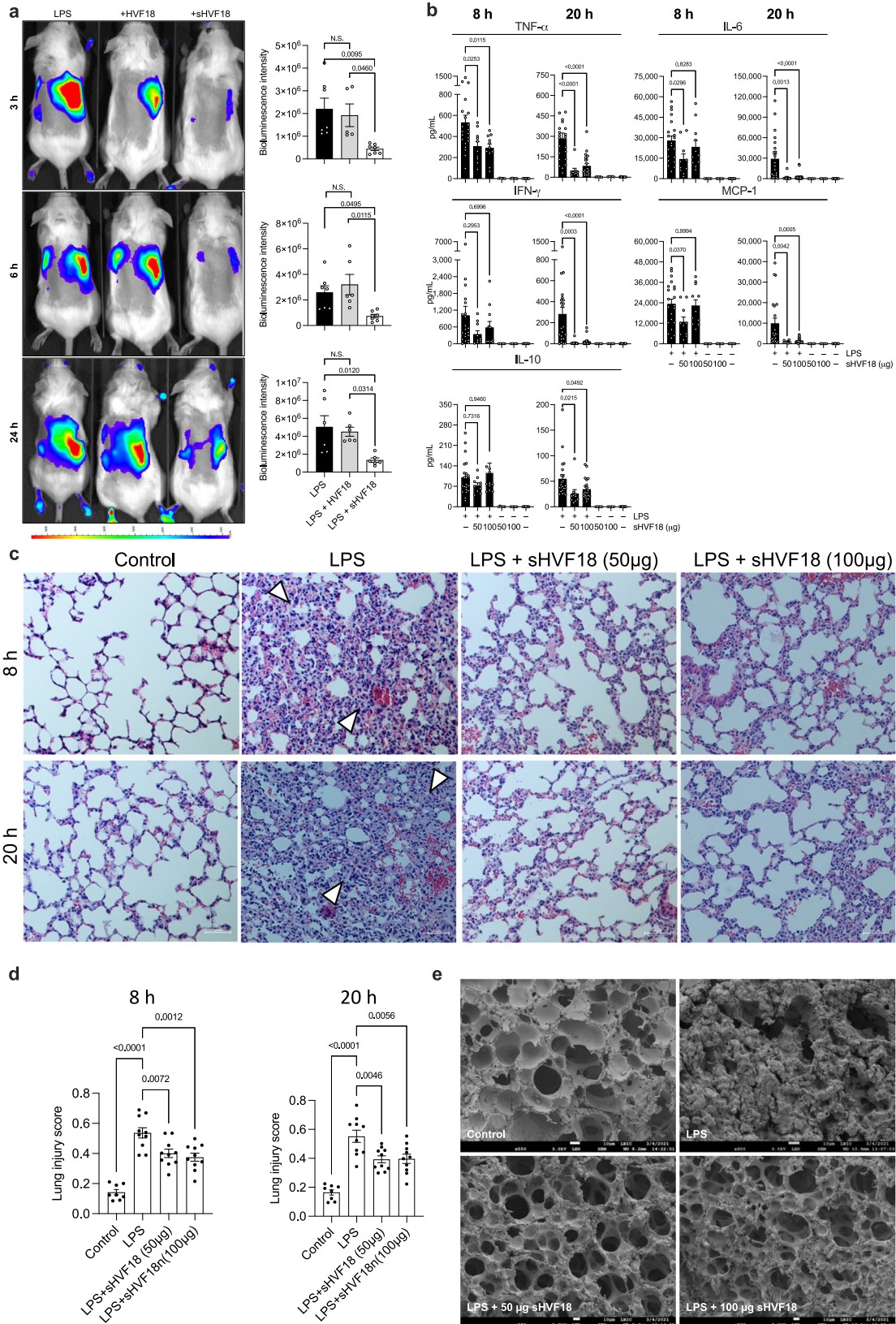

study period of 3–24 h. The linear peptide did not show any inhibitory effects at the same dose (Fig. 4a and Supplementary Fig. 19a). We next used a "classical" systemic model of endotoxin-induced shock. C57BL/6 mice were injected intraperitoneally (i.p.) with a sublethal dose of LPS and treated (i.p.) after 30 min with increasing doses of sHVF18. We observed a significant reduction in plasma TNF-α, IL-6, IFN-γ, and MCP-1

levels by sHVF18 in the dose range 20–100 µg after 20 h (Supplementary Fig. 19b). sHVF18 (50 and 100 µg) and an additional shorter treatment time of 8 h were then selected to study both cytokine and lung changes. Both concentrations of sHVF18 yielded a significant reduction of cytokine levels (Fig. 4b). Moreover, histological analyses of lungs from LPS-treated animals demonstrated a significant inflammatory infiltrate and

**Fig. 4 | Effects of the stapled peptide on endotoxin responses in experimental mouse models. a** A representative in vivo inflammation imaging by IVIS in NF-κB reporter mice. HVF18 or sHVF18 were mixed with LPS immediately before subcutaneous injection on the back of transgenic BALB/c Tg(NF-κB-RE-luc)-Xen reporter mice. In vivo imaging was performed using an IVIS Spectrum bioimaging system at 3–6–24 h after subcutaneous deposition. The bar chart shows the measured bioluminescence intensity emitted from these mice. Data are presented as the means ± SEM; 3h: LPS group ($n = 7$), LPS + HVF18 group ($n = 5$), LPS + sHVF18 group ($n = 7$); 6 h: LPS group ($n = 7$), LPS + HVF18 group ($n = 6$), LPS + sHVF18 group ($n = 7$);. 24 h: LPS group ($n = 6$), LPS + HVF18 group ($n = 6$), LPS + sHVF18 group ($n = 6$). $P$ values were determined using a one-way ANOVA with Tukey's multiple comparisons tests. **b**, Cytokines release from plasma collected after 8 and 20 h from C57BL/6 mice stimulated with a sublethal dose of LPS and then treated with sHVF18. Data for the bar charts are presented as the means ± SEM. For 8 h: LPS group ($n = 21$), LPS + 50 μg sHVF18 group ($n = 10$), LPS + 100 μg sHVF18 group ($n = 11$), 50 μg sHVF18 group ($n = 5$), 100 μg sHVF18 group ($n = 5$), control ($n = 9$); 20 h: LPS group ($n = 22$), LPS + 50 μg sHVF18 group ($n = 10$), LPS + 100 μg sHVF18 group ($n = 22$), 50 μg sHVF18 group ($n = 5$), 100 μg sHVF18 group ($n = 5$), control ($n = 10$). $P$ values were determined using ordinary one-way ANOVA following Dunnett's multiple comparisons tests. **c** Representative images show H&E staining of mouse lung tissue at 8 and 20 h in a mouse model of LPS-induced sepsis. Arrowheads indicate areas of immune cell invasion and hyper-inflammatory condition of the lung tissue. **d** Bar chart depicts lung injury scores based on histological analysis of the lung tissues. Data are presented as the mean ± SEM; control group ($n = 8$), all the other groups ($n = 10$). $P$ values were determined using a one-way ANOVA with Tukey's multiple comparisons test. **e** Representative scanning electron microscopy images showing changes in the lung tissue 20 h after i.p. LPS administration alone or after peptide treatment.

increased pulmonary leakage of proteins and red blood cells, while lungs of sHVF18-treated animals showed significantly less of these LPS-induced pathological effects (Fig. 4c, d, Supplementary Fig. 19c). SEM analysis showed that sHVF18 markedly normalized lung architecture (Fig. 4e) compatible with the histological analysis.

### Effects of sHVF18 on the transcriptome of mice stimulated with LPS

To understand the molecular targets of sHVF18 in vivo, we decided to use the reductionist LPS-induced mouse endotoxemia model described above and examined the effects of 50 μg sHVF18 treatment on the transcriptional response. We performed RNA-seq of lung tissue after 8 and 20 h LPS treatment in the presence or absence of sHVF18. Sample quality was ensured with an even sequencing depth (27.35 ± 1.3 million reads) and a high percentage of reads aligning to the genome (Supplementary Fig. 20a). Samples from mice treated with only buffer (control) showed minimal variability with very few genes differentially expressed between 8 h and 20 h (Supplementary Fig. 20b). LPS induced a strong transcriptional response with 4209 and 5381 genes differentially expressed compared to controls (FDR adjusted $p$-value < 0.01 and |logFC| > 1) after 8 and 20 h, respectively. The most enriched gene ontology (GO) terms included "Interferon response" and "Response to virus" (Fig. 5a). Transcription factor target analysis of significantly enriched genes in LPS-treated animals indicated activation of IRF- and NF-κB driven signatures (Fig. 5b, red ridges). In contrast, the transcriptional response to LPS was markedly blunted in the presence of sHVF18 with only six and 1930 genes differentially expressed 8 and 20 h after co-administration, respectively (compared to 8 h control, Supplementary Fig. 20b). Consistent with this, a core set of inflammatory genes displayed high activity in LPS-treated animals whereas LPS + sHVF18 treated animals were remarkably down-regulated (Fig. 5c) and showed much lower activation of LPS-induced transcription factors (Fig. 5b, blue ridges). Twenty hours after co-administration of LPS, sHVF18 and LPS + sHVF18, a three-way Venn diagram (Fig. 5d) showed that 3971 genes were uniquely regulated by LPS, largely overlapping with the response at 8 h. One-thousand-five-hundred-forty-three were shared between LPS and LPS + sHVF18 and this gene set essentially related to a residual LPS response, with GO-terms such as "Responses to interferon-gamma" and "Positive regulation of cytokine production" being enriched. Finally, genes only regulated in LPS + sHVF18 ($n = 359$) were related to organelle fission, the centromeric chromosome region and humoral immunity, indicating a potential host-modifying response. Peptide alone had a very modest response, with only 31 genes regulated in total with no obvious common theme.

We also wanted to specifically interrogate sepsis-related pathways and downloaded 14 gene sets related to sepsis from the "immunologic signature gene sets" database (MSigDB, version 7.3). In LPS-treated animals (8 h), we saw an upregulation of nine of these 14 gene sets ($p < 0.05$, Supplementary Fig. 21), which was confirmed by plotting the leading-edge genes for these gene sets in a volcano plot (Supplementary Fig. 22a), clearly indicating a strong upregulation of sepsis-related genes. In stark contrast, when we tested these gene sets in LPS + sHVF18 treated animals (8 h), none were significant and plotting the previously identified sepsis genes in a volcano plot (Supplementary Fig. 22b) confirmed minimal regulation of these genes in the presence of sHVF18 + LPS.

### Effects of sHVF18 on survival in an experimental sepsis mouse model

The observation that sHVF18 was able to target multiple sepsis-related genes, such as LCN2, TNIP1, NFkB1 & 2, SNX10, TNFAIP3, MMP14, BCL3, SERPINE1, SOD2, IL6RA, and ICAM1, prompted us to use a more complex in vivo model, i.e., the cecal ligation and puncture (CLP) model of sepsis (Supplementary Fig. 23a). The mouse CLP model is the most widely used model for experimental induction of systemic inflammation during a polymicrobial infection and is currently considered a gold standard in research since it, with certain limitations, mimics the nature and evolution of severe sepsis in humans[36,37]. Usually, CLP-associated lung injury develops within 18 to 72 h. It is characterized by hypoxemia, neutrophilic inflammation, and interstitial and alveolar edema, leading to the deterioration of animal status and mortality. In this sepsis model, intraperitoneal sHVF18 treatment significantly improved animal survival (Supplementary Fig. 23b).

### Effects of sHVF18 in a porcine endotoxin-induced ARDS model

Efficacy studies in large animal models are an important step in pre-clinical drug development and translational studies. We, therefore, investigated the therapeutic effects of sHVF18 in an established pre-clinical porcine model of ARDS[38,39]. ARDS was induced by injecting *E. coli* LPS intravenously (study outlines are presented in Fig. 6a). All pigs developed hemodynamic instability and required inotropic support following LPS administration (inotropic support refers to the use of agents of dobutamine and norepinephrine with the clinical purpose of maintaining hemodynamic stability). This hemodynamic instability and the differences between treated and non-treated pigs over the time course of the experiment are shown by the $PaO_2$ $FiO_2^{-1}$ ratio (Fig. 6b), cardiac output (Fig. 6c), urine output (Fig. 6d), norepinephrine (NA, Fig. 6e), and lactate levels (Fig. 6f). Central venous pressure was low, indicating that the deterioration of the oxygenation capacity was secondary to acute lung injury and not due to *cor pulmonale*.

Following intravenous administration of sHVF18, the treated animals stabilized in their hemodynamics and required significantly less inotropic support compared to the non-treated animals. sHVF18-treated animals did not deteriorate as much as non-treated animals in the $PaO_2$ $FiO_2^{-1}$ ratio (Fig. 6b), and only one of the five treated animals had mild ARDS while the rest did not classify as ARDS (according to Berlin's definition of ARDS[40]). At the endpoint of the experiment, lung tissue samples were taken from the right lower lobe and were

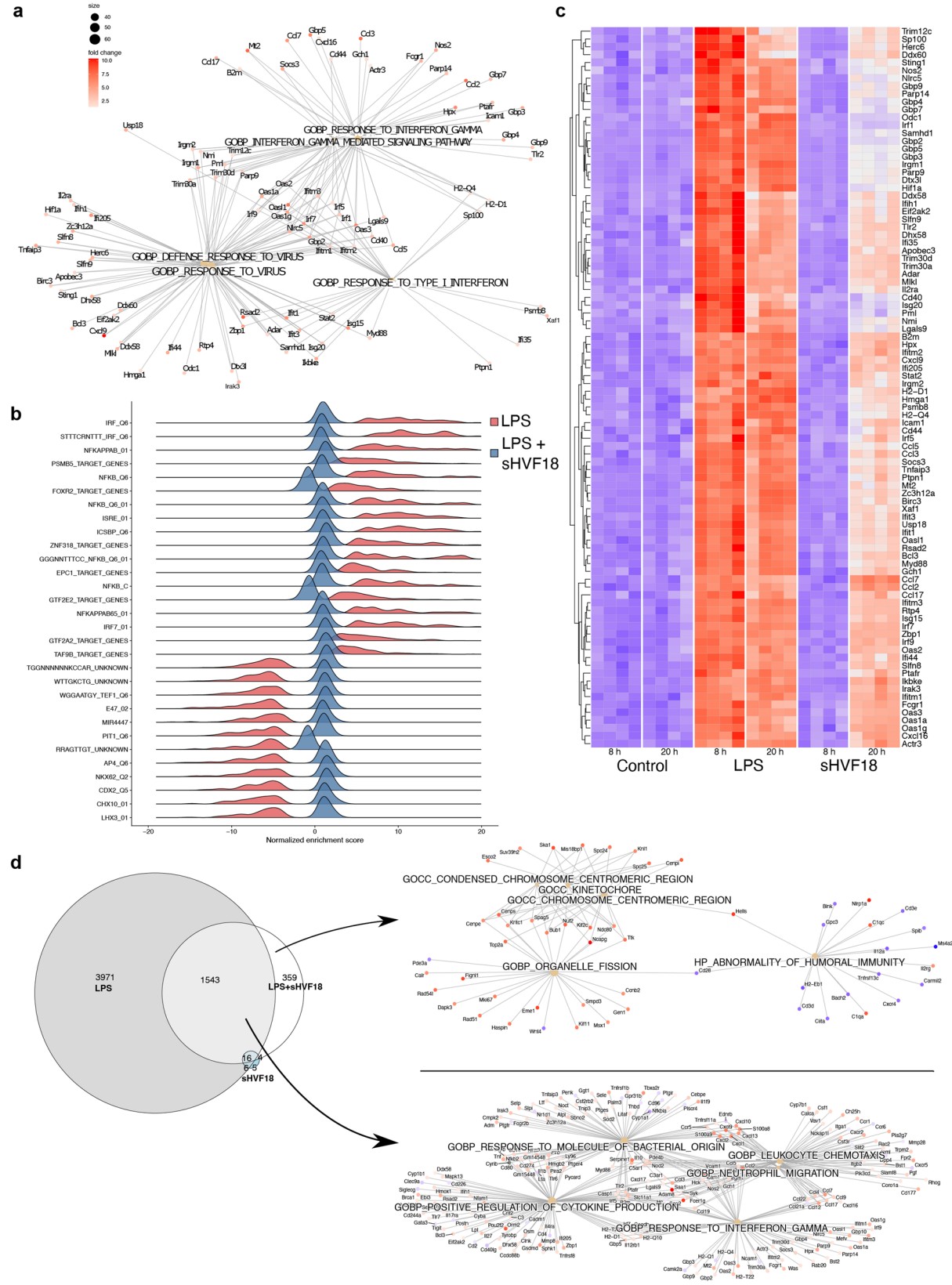

compared to lung tissue samples from five healthy control pigs. Lung biopsies taken from healthy controls for histological analysis appeared normal, with no anomalies (Fig. 6g, left panel). All biopsies taken from both treated and non-treated animals showed infiltration of immune cells and signs of diffuse alveolar damage, including thickening of the alveolar-capillary barrier with intra-alveolar hemorrhage. However,

this was less in the treated pigs (Fig. 6g, middle and right panels). In addition to subjective analysis of lung histology, blinded scoring was performed on all pigs by three independent observers. Significant increases in a cumulative lung injury score in the non-treated group compared to healthy controls were seen, accounting for multiple signs of lung injury following acute lung injury onset via LPS administration.

**Fig. 5 | Effect of sHVF18 on transcriptome profiling in mice stimulated with LPS.**
**a** Linkage plot of genes and enriched GO-terms in LPS-treated animals after 8 h identifies a prototypical LPS response. **b** Ridgeplot showing expression distributions of core enriched transcription factor targets 8 h after LPS treatment (red) and for LPS + sHVF18 (blue) compared to controls (buffer only). IRF and NF-κB are identified as activated transcription factors (positive enrichment score), whereas CDX2 and LHX3 are down-regulated. This regulation is greatly reduced in the presence of sHVF18. **c** Heatmap of core enriched genes identified in lungs from mice treated with buffer only (control), LPS, or LPS + 50 µg mL$^{-1}$, after 8 and 20 h for gene sets identified as differentially regulated. Mice stimulated with LPS showed an upregulation of inflammatory genes, which are abrogated by sHVF18 treatment at 8 h and, to a lesser extent, at 20 h. **d** Venn diagram of the number of overlapping genes differentially expressed (P value < 0.01 and |logFC| > 1 were adjusted for multiple comparisons using the Benjami-Hochberg method) for LPS, LPS + sHVF18, or sHVF18 alone compared to control at 20 h. P values was calculated using a linear two-sided model. Differentially expressed genes were further analyzed for over-enriched GO terms and presented as linkage plots as in **a**.

Significant fewer signs of lung injury were seen in the treated pigs compared to the non-treated pigs (Fig. 6h).

## Discussion

Infectious diseases and systemic complications such as sepsis account for millions of deaths worldwide each year. The decreasing effectiveness of antibiotics and other antimicrobial agents because of resistance development is an increasing problem. The need for innovative and effective treatments is therefore urgent. Current drug lead development has typically focused on developing drug candidates characterized by a single target, usually with high, nanomolar, affinities. For this, screening of vast amounts of chemical libraries has often been used, in combination with structure activity-based drug design in iterative processes in order to generate suitable high affinity compounds with the desired mode of action and low toxicity[41]. An example is TAK 242 (resatorvid), a small-molecule inhibitor of TLR4 signaling, initially developed as an anti-sepsis agent specifically disrupting the TLR4-TIRAP interaction with high selectivity and capable of inhibiting inflammatory mediator production at nanomolar levels[42–44]. Moreover, in order to speed up the drug development process and reduce the costs and time for clinical studies, drug repurposing has been employed[45], resulting in drug candidates such as simvastatin[46] and several oncology drugs[47].

However, finding drug candidates that bind single targets selectively and at high affinity has been questioned, suggesting it is far more productive to aim for several targets at the same time (the multi-target, polypharmacological drug approach)[48,49]. Interestingly, biological systems are generally governed by a multitude of weak and/or transient interactions (dissociation constant: $K_d$ > µM), either working alone or in concert[50]. It is therefore notable that the evolutionarily conserved TCP innate fold, containing the sequence KKWIQK, participates not only in specific CD14 binding at micromolar levels, but also interacts with LPS. Moreover, TCPs modulate responses to other bacterial products such as lipoteichoic acid, peptidoglycan, and bacterial DNA[27], inhibit contact activation[17], and dampen neutrophil chemotaxis[18], illustrating a promiscuous interaction with various targets. TCPs can therefore be viewed as natural "transient drugs", having a high off/low on rate, multivalency, and multiple targets[15,50]. The transient drug action can indeed explain the observation that TCPs down-modulate excessive responses to bacteria and bacterial products such as LPS, without completely abrogating inflammation, as it also has a physiologically relevant function in infection clearance and wound healing. Indeed, such phenomena can explain the therapeutic effects in experimental animal models of systemic bacterial infection and endotoxic shock, where treatment with TCPs reduced cytokine responses and animal mortality[14,17] as well as in wound infection models, where TCP GKY25 cleared wound infection, normalizing healing[19,20].

The major achievement in this work is therefore the successful design of a peptide-based mimetic of the TCP innate fold, providing a drug class based on Nature's own anti-infective strategies. From a drug design perspective, we demonstrate that an approach based on a combination of structure-based design, evolutionary considerations, and molecular simulations is effective in generating several promising drug leads. The finding that peptide variants down to 13 amino acids showed anti-inflammatory effects, confirmed the dependence of the structural innate TCP fold for drug activity. From this generated set, the optimal sHVF18 was selected based on a favorable therapeutic index. The mode of action of the designed sHVF18 was demonstrated using a broad arsenal of biochemical, biophysical, in silico, cellular and in vivo studies. Microscale thermophoresis, HDX-MS, and computational modeling showed that sHVF18 binds with high affinity to the N-terminal hydrophobic LPS binding pocket of human CD14. Indeed, HDX-MS data elegantly confirmed the in silico predictions for CD14 binding, both methods identifying the LPS-binding N-terminal region of CD14 as a target for sHVF18. This is in agreement with recent reports describing the pathway for the transfer of lipid A from CD14 to TLR4/MD-2[51], which highlights that sHVF18s binding near the hydrophobic cavity could either impede the transfer of the lipid or occlude the interface between the receptor proteins. The dose needed for sHVF18 to achieve these effects is much lower than for the original HVF18, illustrating the successful development of sHVF18 as a drug concept for systemic use. Moreover, while the endogenous HVF18 sequence is proteolytically degraded, sHVF18 showed increased resistance to proteases[52]. As proteolysis is a hallmark of systemic infection and sepsis[53], reduced degradation of sHVF18 in a highly proteolytic environment during infection may hence further improve the pharmacological profile and reduce peptide dose and treatment frequency.

Many drugs in preclinical development that show a promising target profile and toxicity in vitro fail in later in vivo development stages. It is therefore notable that the therapeutic potential of sHVF18 is supported by experimental data from both small and large animal models. LPS-induced microvascular lung injury in mice is a common experimental model reflecting ARDS development during sepsis[54]. In mice subjected to LPS-induced shock, we found that LPS-induced upregulation of multiple sepsis-related genes in lung tissue and cytokine levels in plasma are reduced by sHVF18 treatment. It is particularly notable that the peptide shows a broad target profile, down-modulating multiple sepsis-related pathways. Moreover, sHVF18 normalizes lung tissue architecture with visibly less oedema and fibrin deposits in the alveolar spaces, compatible with a reduction of inflammation induced coagulation, as previously described for the endogenous TCP GKY25[17]. In contrast to the LPS-model, the mouse CLP model is more complex, yielding systemic inflammation triggered by fecal Gram-positive and Gram-negative bacteria. The fact that sHVF18 treatment significantly improves animal survival is therefore compatible with the peptide's broad targeting of bacterial products from these bacterial groups along with inhibition of CD14.

Hemodynamic derangements and lung damage is a hallmark of human sepsis, and the use of the porcine LPS model of ARDS[38,39] was therefore employed to enable an important translation to human studies. In this model, the excessive systemic inflammation is expressed as a sudden decrease in blood pressure, need for inotropic support, and reduced blood oxygenation and urine production. Analysis of lung biopsies confirmed the presence of acute lung injury, consistent with ARDS. sHVF18-treated pigs had an overall better outcome and became more hemodynamically stable over time with a reduced need for adrenaline after sHVF18 administration, and blood oxygenation and urine output were increased. sHVF18 thus ameliorated the symptoms of systemic inflammation and hindered the development of severe states of ARDS in this pig model.

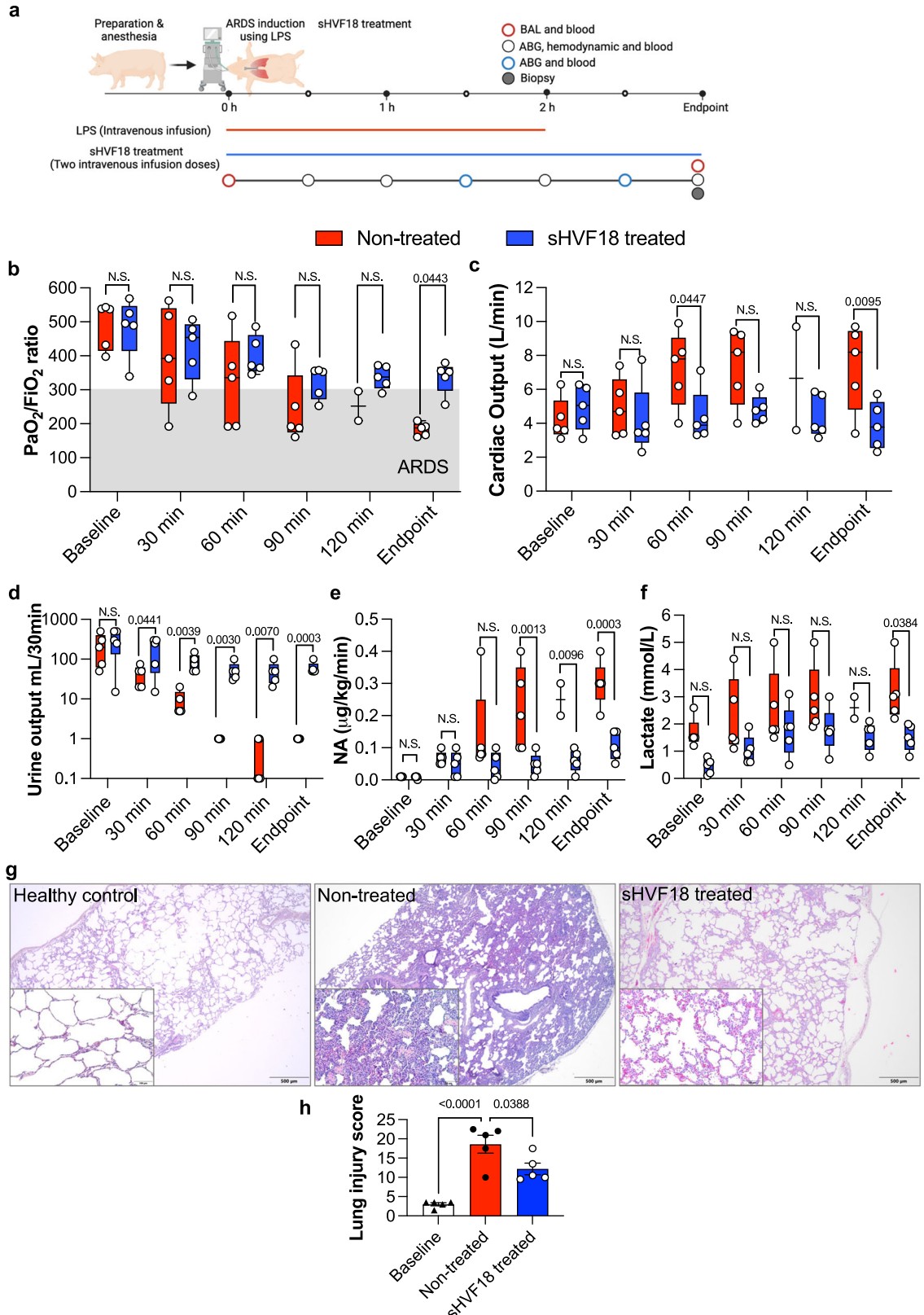

Collectively, these results demonstrate a successful approach of transforming an innate structural fold into a peptide-based drug targeting TLR-driven systemic inflammation and moreover, show the therapeutic potential of sHVF18 in the reductionist endotoxin model and the more complex systemic polymicrobial infection model in mice. The marked therapeutic effects in the porcine LPS-induced ARDS model further indicate the therapeutic potential of the transient drug concept based on TCP´s innate fold.

While the herein described in vivo models offer valuable preclinical data, translating these findings to human studies on infected patients presents challenges. Animal models, while necessary and important in preclinical development, may not fully replicate the

**Fig. 6 | Intravenous administration of sHVF18 ameliorated symptoms of systemic inflammation and hindered the development of severe states of ARDS in pigs.** The figure shows results from pigs with acute lung injury and ARDS treated with and without sHVF18. **a** Schematic overview of the experiment setup. Pigs were anesthetized in mechanical ventilation and monitored continuously using an arterial line and Swan-Ganz catheter. LPS was given intravenously, and treated animals received sHVF18. Figure created with BioRender.com. **b**–**f** Hemodynamics, vitals, and pulmonary gas exchange were followed continuously over the time course of the experiment ($n = 5$ for each group, except for not-treated at 120 minutes, $n = 2$). **b** Pulmonary gas exchange as $PaO_2 \, FiO_2^{-1}$ ratio between treated and non-treated pigs. All non-treated animals developed mild to moderate ARDS. A significant increase in cardiac output (**c**), a significant decrease in urine output (**d**), and a significantly increased need for inotropic support (**e**), and a significant increase in lactate levels (**f**) were seen in the non-treated animals but not in the treated animals indicating a severe stage of systemic inflammation and organ dysfunction in the non-treated pigs. The box in the figures (**b**–**f**) extends from the 25th to 75th percentiles. Whiskers represents minimum to maximum values. The line in the middle of the box is plotted at the median. **g** Images representative of $n = 15$ samples of hematoxylin and eosin (H&E) histology of healthy controls ($n = 5$) (left), non-treated (middle) ($n = 5$) and treated (right) lungs ($n = 5$). The scale bar in the larger image represents 0.5 mm. The callout shows a magnified portion of the tissue where the scale bar represents 0.2 mm. **h** The results of cumulative blinded scoring of the histology ($n = 5$ in each group). Statistically significant differences between non-treated and treated groups were tested with a two-sided Student's $t$-test (**d**), within groups with two-way ANOVA with Šidak´s multiple comparisons test (**b**, **c**, **e** and **f**) or with one-way ANOVA with Tukey's multiple comparisons test (**h**).

complexity of human bacterial infections and the development of systemic inflammation in various sepsis endotypes. Furthermore, the variability in human patient populations, including genetic backgrounds, coexisting conditions, and compromised immune systems, can impact treatment response and drug efficacy. The timing of peptide treatment and the stage of sepsis development are also crucial factors affecting therapeutic interventions. To address these limitations, it is essential to investigate variables such as peptide pharmacokinetics, timing of dosing, metabolism, and toxicity, and characterize treatment effects in infection models that represent selected sepsis endotypes. This comprehensive approach will help overcome these challenges and determine whether sHVF18, with its multimodal and broad action, can provide a therapeutic opportunity for effectively targeting excessive systemic inflammation and ARDS during bacterial infections in patients.

## Methods

### Study design
The goal of this study was to use structural cues of the innate structural fold of endogenous TCPs for design and development of a peptide-based drug targeting TLR-driven excessive inflammation in systemic infection and sepsis. The approach utilized a combination of structure-based design, evolutionary considerations, and molecular simulations to generate several promising drug leads. The generated leads were then analyzed for hemolysis, and anti-inflammatory activity in cell models and human blood, and the optimal sHVF18 was selected based on a favorable therapeutic index. Protease resistance was assessed using gel electrophoresis analysis and mass spectrometry, and the structure of sHVF18 was elucidated using nuclear magnetic resonance analysis, HDX-MS, and circular dichroism spectroscopy. The interaction with the target receptor CD14 and its hydrophobic LPS-binding pocket was assessed using a combination of microscale thermophoresis, HDX-MS, and computational modeling. sHVF18 efficacy in vivo was demonstrated using local subcutaneous LPS-induced inflammation in NF-κB reporter mice and endotoxin-induced shock in C57BL/6 mice. The systemic endotoxin model was utilized to study the transcriptional response and define inflammatory pathways targeted by sHVF18 treatment. Effect of sHVF18 on animal survival was shown in the complex cecal ligation and puncture (CLP) model of sepsis. Since hemodynamic and kidney derangements, as well as lung damage, are hallmark features of human sepsis, these parameters were studied in a porcine LPS model of septicemia-induced ARDS to provide preclinical efficacy data that enables translation to human studies.

### Ethics statement
Venous blood was collected from healthy donors after written informed consent was obtained. After collection, whole blood or its fraction, such as plasma and serum, were used immediately or stored at −80 °C. The use of blood was approved by the Ethics Committee at Lund University, Lund, Sweden (permit number: DNR2015/801). All experiments on mice were performed according to Swedish Animal Welfare Act SFS 1988:534 and were approved by the Animal Ethics Committee of Malmö/Lund, Sweden (permit numbers M8871-19, 16542-21). Mice were housed in Innovive IVC Rodent Caging System on a 12/12 h light/dark cycle. The ambient temperature ranged between 19 and 23 °C with humidity 55 ± 10%. All animal had free access to water and chow. The study on pigs was approved by the local Ethics Committee for Animal Research (DNR 5.2.18-4903/16, and DNR 5.2.18-8927/16) at Lund University. All animals received care according to the USA Principles of Laboratory Animal Care of the National Society for Medical Research, Guide for the Care and Use of Laboratory Animals, National Academies Press (1996).

### Peptides and proteins
The peptides GKY25 (GKYGFYTHVFRLKKWIQKVIDQFGE), HVF18 (HVFRLKKWIQKVIDQFGE), and their respective stapled versions denoted as sGKY25 (GKYGFYTHVFRLKKWIXKVIXQFGE) and sHVF18 (HVFRLKKWIXKVIXQFGE), shorter version of sHVF18 (denoted as sVFR17, sFRL16, sRLK15, sLKK14, sKKW13), and stapled GKY25 with a cyclized N-terminal region (cyclo(GKYGE)YTHVFRLKKWIXKVIXQFGE) denoted as 2sGKY25, were synthesized by AmbioPharm, Inc. (North Augusta, SC, USA). Briefly, standard 9-fluorenylmethyloxycarbonyl (Fmoc) solid-phase peptide synthesis (SPSS) was used. To obtain hydrocarbon stapled peptides, olefin-bearing (S)-2-(4'pentenyl)-alanine was inserted at specified locations in the respective peptide sequences (denoted by X)[24]. Olefin metathesis reaction was performed on a solid support using Grubbs' first-generation catalyst in 1,2-dichloroethane. To generate 2sGKY25, F5 was replaced by E5, enabling covalent coupling between positions 1 and 5 via amide formation and a lactam bridge. The product peptides were cleaved from the resin and further purified by RP-HPLC. Peptides were provided as acetate salts, and the purity was confirmed with MALDI-TOF MS (>95%). Human His-tag CD14 (hCD14) was produced in insect cells and purified, as reported previously[55]. Deglycosylated hCD14 was obtained using the PNGase F Glycan Cleavage Kit (Thermo Fisher Scientific, Carlsbad, CA, USA) following the manufacturer´s protocol.

### Circular dichroism spectroscopy
The secondary structure of GKY25, HVF18, and their respective stapled versions, with and without LPS from *Escherichia coli O111:B4* (LPS, Sigma-Aldrich, Saint Louis, MO, USA), were assessed by circular dichroism (CD). The peptides were diluted to 10 µM in 10 mM Tris at pH 7.4 and incubated with 100 µg mL$^{-1}$ LPS for 30 min at 37 °C. A Jasco J-810 spectropolarimeter (Jasco, Tokyo, Japan) equipped with a Jasco CDF-426S Peltier set to 25 °C was used to perform the measurements. The spectra were recorded between 190 and 260 nm (scan speed: 20 nm min$^{-1}$) as an average of 5 measurements in a 0.2-cm quartz cuvette (Hellma, GmbH & Co, KG, Müllheim, Germany). The baseline (10 mM Tris at pH 7.4 ± 100 µg mL$^{-1}$ LPS) was subtracted from each spectrum, and the final signal was converted to mean residue ellipticity, θ (mdeg cm$^2$ dmol$^{-1}$) as reported by Morrissette et al.[56].

In another set of experiments, linear and stapled versions of HVF18 were mixed with 25 or 50% TFE or in water with 100 μg mL⁻¹ LPS, then spectra were acquired as reported above.

## RP-HPLC

HVF18 and its stapled version (2.5 μg) were analyzed on a reverse-phase C18-column (Kinetex 50 × 2.1 mm 2.6 μM, 100 Å pore size, Phenomenex, Torrance, CA, USA) coupled to an Agilent 1260 Infinity System according to previous protocols[57]. Briefly, the column was equilibrated using 95% of buffer A containing 0.25% of TFA in Milli-Q water and 5% of Buffer B containing 0.25% of TFA in acetonitrile. The peptide was pre-mixed with Buffer A (1:3) 5 min before being loaded on the column. Protease-digested peptides were analyzed by HPLC as described above. Samples from two different digestions were analyzed.

## Protease degradation of peptides

Peptides were resuspended in endotoxin-free water at a final concentration of 1 mM. Twenty micrograms of GKY25 and sGKY25 or 14.7 μg of HVF18 and sHVF18 were incubated with 0.2 μg of human neutrophil elastase (HNE, Calbiochem®, Merck, Darmstadt, Germany), *P. aeruginosa* elastase (PE, Calbiochem®, Merck, Darmstadt, Germany), Glutamyl-C endopeptidase (EC 3.4.2.11.9) from *S. aureus* V8 (V8, BioCol GmbH, Michendorf, Germany), or trypsin (Try, Promega, Madison, WI, USA) for different lengths of time (0–18 h) in a final volume of 20 μL. The material was analyzed by SDS-PAGE and LC-MS/MS (for GKY25, sGKY25, HVF18 and sHVF18) and by RP-HPLC (for HVF18 and sHVF18). All digestions were performed in three independent experiments.

## SDS-PAGE

Two micrograms of peptide from each condition were loaded on 10–20% Novex Tricine pre-cast gels (Invitrogen, Carlsbad, CA, USA). The run was performed at 100 V for 100 min. The gel was stained by using Coomassie Brilliant blue (Invitrogen, Carlsbad, CA, USA), and images were acquired with a Gel Doc Imager (Bio-Rad Laboratories, Hercules, CA, USA). Samples from three independent digestions were analyzed.

## LC-MS/MS

LC-MS/MS analysis was performed using an HFX Orbitrap MS system (Thermo Scientific, San Jose, CA, USA) equipped with a Dionnex 3000 Ultimate HPLC (Thermo Fisher, USA). Settings were used as previously described by Petruk et al.[55] with minor changes. MS range was 350–1600 m/z, the start of LC gradient at 4 min, solvent B was increased from 4% to 10% during 3 min, to 65% during 11 min, followed by 100% for 2 min. MS/MS spectra were searched with PEAKS (version 10) against UniProt Homo Sapiens (version 2020_02). The precursor tolerance was set to 10 ppm and 0.02 Da for the MSMS fragments, and no enzyme was selected, allowing unspecific cleavage. Samples from two independent digestions were analyzed.

## NF-κB activation assay

Effects on NF-κB activation were assessed using THP1-XBlue-CD14 reporter cells (InvivoGen, San Diego, CA, USA). Briefly, 180,000 cells well⁻¹ were seeded in 96-well plates in phenol red RPMI media, supplemented with 10% (vv⁻¹) heat-inactivated FBS and 1% (vv⁻¹) Antibiotic-Antimycotic solution (AA). *E. coli* LPS (100 ng mL⁻¹, Sigma Aldrich, Saint Louis, MO, USA) was then added with and without the peptides GKY25, sGKY25, HVF18, and sHVF18 at different concentrations (1–20 μM). NF-κB activation was determined after 20 h of incubation according to the manufacturer's instructions (InvivoGen, San Diego, CA, USA) by mixing 20 μL of supernatant with 180 μL of SEAP detection reagent (Quanti-BlueTM, InvivoGen, San Diego, CA, USA), followed by absorbance measurement at 600 nm. Data shown are mean values ±

SD obtained from at least four independent experiments, all made in triplicate. In another set of experiments, the cells were stimulated with 100 ng mL⁻¹ of *E. coli* LPS, 1 μg mL⁻¹ *S. aureus* LTA, 1 μg mL⁻¹ *E. coli* PGN, 1 μg mL⁻¹ *S. aureus* PGN, or 10 μg mL⁻¹ *S. cerevisiae* zymosan in the presence or the absence of 10 μM of linear and stapled HVF18. The experiment was performed as described above.

## Hemolysis assay

Fresh venous blood from healthy donors was collected in tubes containing lepirudine (50 μg mL⁻¹). Then, 50 μL of blood was transferred to a round-bottom 96-wells plate containing 150 μL of peptides previously diluted in RPMI-1640- GlutaMAX-I without phenol red (Gibco, Thermo Scientific, San Jose, CA, USA). Diluted (1:4) blood was used as a negative control. Blood (50 μL) mixed with 150 μl 5% Tween-20 was used as a positive control. After 1 h incubation at 37 °C and 5% CO₂, the plate was centrifuged at 800 g, 150 μL of each sample was transferred to a flat-bottom 96-wells plate, and the absorbance at 450 nm was measured. The percentage of hemolysis was calculated following the formula (1) reported below:

$$\% \text{Hemolysis} = \frac{\text{Abs } 450\,\text{nm (Sample} - \text{control)}}{\text{Abs } 450\,\text{nm (Posivite control} - \text{control)}} \times 100 \quad (1)$$

To evaluate the hemolytic effect of the peptides on erythrocytes, blood was collected as reported above and centrifuged at 250 g for 10 min, plasma was discarded, and red blood cells were washed with 150 mM NaCl, 10 mM Tris, pH 7.4 for three times. The pellet was diluted 100 times in 150 mM NaCl, 10 mM Tris, pH 7.4. One-hundred microliters of this solution were added to a round-bottom 96-well plate containing 100 μL of peptides. After 1 h incubation at 37 °C and 5% CO₂, the plate was centrifuged, and the absorbance at 450 nm was measured. The percentage of erythrocyte lysis was determined as above.

## Whole blood LPS stimulation assay

Fresh venous blood from healthy donors was collected in the presence of lepirudin (50 μg mL⁻¹). The blood was diluted 1:4 in RPMI-1640-GlutaMAX-I (Gibco), and 1 mL of this solution was transferred to 24-well plates and stimulated with 100 ng mL⁻¹ of LPS immediately after adding increasing (1–10 μM) peptide concentrations. After 24 h incubation at 37 °C in 5% CO₂, the plate was centrifuged for 5 min at 1000 × g, and then the supernatants were collected and stored at −80 °C before analysis. The experiment was performed at least four times by using blood from different donors each time.

## Cytokine assay

Plasma obtained from the blood experiment was used to evaluate cytokine release. A Human inflammation DuoSet® ELISA Kit (R&D Systems, Minneapolis, MN, USA) specific for TNF-α and IL-1β was used according to the manufacturer's instructions. Absorbance was measured at a wavelength of 450 nm. Data shown are mean values ± SEM obtained from at least four independent experiments, all performed in duplicate. TNF- α, IFN-γ, MCP-1, IL-10, and IL-6 in murine plasma were assessed using the Mouse Inflammation Kit (Becton Dickinson AB) according to the manufacturer's instructions.

## NMR spectroscopy

Data acquisition: All NMR experiments were performed on a 700 MHz Bruker Avance III HD spectrometer (Swedish NMR Centre, Gothenburg, Sweden) equipped with QCI cryo-probe and pulse field gradients. Four samples were prepared for NMR: 0.5 (a) and 2.6 mM (b) sHVF18 in DMSO-d6 supplemented with 4,4 dimethyl-4-silapentane-5-sulfonate sodium salt (DSS) to a concentration of 100 μM; 1.6 mM sHVF18 in 25% (c) or 50% (d) 2,2,2-Trifluoroethanol (TFE), supplemented with 10% D₂O, 200 μM DSS, 0.02 vol% NaN₃ at pH 4.5. ¹H spectra were acquired from all samples at 298 K. Based on the ¹H

spectra and TOCSY spectra (80 ms mixing time), a series of TOCSY (40 and 80 ms mixing time), NOESY (100 and 150 ms mixing time), ROESY (100 and 150 ms mixing time), DQF-COSY, $^{13}$C-edited HSQC, $^{15}$N-SOFAST-HMQC, $^{13}$C HSQCTOCSY and $^{13}$C-HMBC spectra were acquired on the sample (b) and (d). Spectra were processed using nmrPipe with squared cosine apodization and zero-filling in both dimensions. Spectra were analyzed and assigned using CCPNMR v2.4. Spin systems were identified using a combination of NOESY and TOCSY spectra, where NOESY cross-peaks were used to assign inter-residue connections.

Structure calculation: The 3D structure of the stapled peptide sHVF18 was determined using distance and dihedral angle restraints from assigned NOE cross-peaks and chemical shifts obtained for sample (d) described in NMR spectroscopy. Distance restraints from NOE cross-peaks for sHVF18 were generated and calibrated using the CCPNMR suite[58]. Only NOE cross-peaks with unambiguous assignments were considered for the structure calculations. Dihedral angle restraints were generated using the DANGLE software in the CCPNMR software suite, where dihedral angle restraints are generated based on chemical shift data from backbone atoms. Initially, extended peptide structures of sHVF18 were generated using in-house software. The initial structure was energy-minimized and equilibrated in a vacuum. For the amino acids constituting the stapled peptide, the amino acids were generated as separate non-natural amino acids and parametrized for GROMACS using ACPYPE[59]. After energy minimization, bond restraints were applied for the Ce of each amino acid of the staple and the covalent bond was generated using inbuilt functions in GROMACS. Structure calculations were made using the GROMACS molecular dynamics suite[60] using the AMBER99SB force field. Solvation of sHVF18 was done using 1594 water atoms with TIP3P topology. The charge of the system was set to zero by adding three chloride atoms to the system. The initial stapled peptide structure was prepared for structure calculations by energy minimization followed by short equilibration simulations at constant volume or pressure. Structure calculations were then performed in an iterative fashion, where 15 structures of the peptide were calculated using distance and dihedral angle restraints and then evaluated using the CCPNMR suite, where restraints were reassigned or removed. Restraints with correct assignment upper bond violations were iteratively recalibrated by applying a soft upper bond restraint. This was repeated until a converged structure ensemble was reached, of which ten structures out of 60 were chosen as a representative ensemble based on Ramachandran backbone dihedral angles, distance restraint violations and a converged backbone RMSD. The structure quality of the ensemble structure was evaluated using in-house software and the MolProbity software suite[61].

The NMR data have been deposited to the wwPDB. The PDB code for this deposition is 8BWW. The PDB DOI for this deposition is 10.2210/pdb8bww/pdb. The BMRB code for this deposition is 34778.

## Hydrogen-deuterium exchange mass spectrometry (HDX-MS)

All chemicals were from Sigma-Aldrich, and pH measurements were made using a SevenCompact pH meter equipped with an InLab Microelectrode (Mettler-Toledo) after four-point calibration (pH 2, 4, 7, and 10). HDX-MS analysis was performed using automated sample preparation on a LEAP H/D-X PAL™ platform interfaced to an LC-MS system, comprising an Ultimate 3000 micro-LC coupled to an Orbitrap Q Exactive Plus MS. The samples comprised recombinant deglycosylated CD14 (2.4 mg mL$^{-1}$) in 20 mM Tris, 100 mM NaCl, 10% glycerol, at pH 7.5 with and without ligands, HVF18, sHVF18, LPS. The peptides and LPS were prepared fresh in milli-Q water using short cycles of ultrasonication. The interaction samples were prepared to have a molar ratio of 1:10 (CD14:ligands), assuming a molecular weight of 2.2 kDa for peptides and 10 kDa for LPS.

Unbound/Apo state CD14 samples constituted 2 μL and 2 μL Milli-Q water and interaction analysis samples, 2 μL CD14 mixed with 2 μL ligand, samples were diluted with 26 μL 25 mM Tris, pH$_{(read)}$ 7.5 or HDX labeling buffer of the same composition prepared in D$_2$O, pH$_{(read)}$ 7.1. HDX labeling was carried out for t = 0, 30, 300, 3000 and 9000 s at 20 °C. The labeling reaction was quenched by dilution of 25 μL labeled sample with 30 μL of 1% TFA, 0.4 M TCEP, 4 M urea, pH 2.5 at 1 °C, 50 μL of the quenched sample was directly injected and subjected to online pepsin digestion at 4 °C (in-house made immobilized pepsin column, 2.1 × 30 mm). The online digestion and trapping were performed for 4 min using a flow of 50 μL min$^{-1}$ 0.1% formic acid, pH 2.5. The peptides generated by pepsin digestion were subjected to online SPE on a Pep-Map300 C18 trap column (1 × 15 mm) and washed with 0.1% FA for 60 s. Thereafter, the trap column was switched in line with a reversed-phase analytical column, Hypersil GOLD, particle size 1.9 μm, 1 × 50 mm, and separation was performed at 1 °C using a gradient of 5–50% B over 8 min, and then from 50 to 90% B for 5 min, the mobile phases were 0.1% formic acid (A) and 95% acetonitrile/0.1% formic acid (B). Following the separation, the trap and column were equilibrated at 5% organic content until the next injection. The needle port and sample loop were cleaned three times after each injection with mobile phase 5%MeOH/0.1%FA, followed by 90% MeOH/0.1%FA and a final wash of 5%MeOH/0.1%FA. After each sample and blank injection, the pepsin column was washed by injecting 90 μL of pepsin wash solution 1% FA /4 M urea /5% MeOH. A full blank was run between each sample injection. Separated peptides were analyzed on a Q Exactive Plus MS, equipped with a HESI source operated at a capillary temperature of 250 °C with sheath gas 12, Aux gas 2, and sweep gas 1 (au). Data-dependent MS/MS HCD was used for the identification of pepsin-generated peptides. For HDX analysis, MS full scan spectra at a setting of 70 K resolution, AGC 3e6, Max IT 200 ms and scan range 300–2000 were collected.

## HDX-MS data analysis

PEAKS Studio X Bioinformatics Solutions Inc. (BSI, Waterloo, Canada) was used for peptide identification after pepsin digestion of undeuterated samples (i.e., timepoint 0 s.). The search was done on a FASTA file with only the CD14 sequence; the search criteria were a mass error tolerance of 15 ppm and a fragment mass error tolerance of 0.05 Da, allowing for fully unspecific cleavage by pepsin. Peptides identified by PEAKS with a peptide score value of log $P > 25$ and no modifications were used to generate a peptide list containing peptide sequence, charge state and retention time for the HDX analysis. HDX data analysis and visualization were performed using HDExaminer, version 3.3.0 (Sierra Analytics Inc., Modesto, US). The analysis allowed only for EX2, and the two first residues of a peptide were assumed unable to hold deuteration. As a full deuteration experiment was not done, the full deuteration was set to 75% of max uptake with no back exchange correction, hence reported uptake values are relative. The presented deuteration data is the average of all high and medium confidence results. The allowed peptide retention time window was ±0.5 min. Heatmap settings were uncolored proline, no smoothing and the difference heatmaps were drawn using the volcano plot significance lines settings (see Supplementary Data 1). Information regarding experimental details and significance, as well as uptake and peptide map overlap, is shown for respective states in Supplementary Data 1.

The MS raw files have been deposited to the ProteomeXchange Consortium via the MassIVE partner repository (reference ID: MSV000090815, [https://massive.ucsd.edu/ProteoSAFe/dataset.jsp?task=20d9f7e7f6434abfa3d56e0bf4eedf99]). The MassIVE DOI for this deposition is 10.25345/C5KD1QQ7S.

## Microscale thermophoresis

Microscale thermophoresis (MST) was performed on a NanoTemper Monolith NT.115 apparatus (Nano Temper Technologies, Germany). A

Monolith NT Protein labeling kit RED – NHS (Nano Temper Technologies, Germany) was used to label 20 μM of recombinant hCD14 according to the manufacturer's protocol. Glycosylated hCD14 (5 μL of 21 nM) was incubated with increasing concentrations of GKY25, HVF18 or sHVF18 (0.03–1000 μM) in 10 mM Tris at pH 7.4 with or without 150 mM NaCl in a ratio of 1:1. The sample was then loaded into standard glass capillaries (Monolith NT Capillaries, Nano Temper Technologies), and MST analysis was performed (settings for the light-emitting diode and infrared laser were 80%). The results shown are mean values ± SD of six measurements.

In another set of experiments, LPS-FITC (5 μg mL$^{-1}$; Sigma) was incubated with increasing concentrations of HVF18 or sHVF18 (0.015–500 μM) in 10 mM Tris at pH 7.4 in a ratio of 1:1. This was carried out as described above. The results shown are mean values ± SD of four measurements.

### Integrative modeling of human CD14

The structure of the soluble domain of human CD14 was based on the X-ray structure of human CD14 (PDB: 4GLP)[2]. The missing N-terminal region (residue 20–25) and disulfide bridge (C25-C36) were modeled based on the crystal structure of mouse CD14 (PDB: 1WWL)[3] using Modeller version 9.21[4]. Ten models were constructed, from which three were chosen based on having the lowest discreet optimized protein energy (DOPE) score[5] for the further stereochemical assessment using Ramachandran analysis[6]. The model with the least outlier residues was selected for subsequent docking and simulations.

### In silico analysis of peptide stapling positions

The NMR structure of HVF18 (PDB: 5Z5X)[15] was docked onto the modeled structure of human CD14 using the ClusPro web server[62]. Similar results were obtained as in our previous study, whereby the peptide binds favorably to the N-terminal pocket of CD14[15]. The structure of the top-scoring docking pose was selected as a template for model GKY25. The N-terminal GKYGFYT residues were modeled using Modeller version 9.21, and the model with the lowest discrete optimized protein energy score[63] was chosen.

The GKY25-CD14 complex was solvated with TIP3P water and 0.15 M NaCl salt using the CHARMM-GUI Solution Builder[64]. The steepest descent energy minimization and a 125 ps equilibration simulation, whereby positional restraints with force constant (Fc) of 400 kJ mol$^{-1}$ nm$^{-2}$ were applied to the backbone atoms of the protein and peptide, was performed following the standard CHARMM-GUI protocols[65]. The final snapshot after equilibration was extracted, and the binding energy between GKY25 and CD14 was estimated using the Molecular mechanics Poisson−Boltzmann surface area (MMPBSA) method[66]. A hydrophobic staple was added to the GKY25 peptide by mutating residues at positions i and i + 3 to alanine and linking them with two pentene segments using the CHARMM-GUI Solution Builder[64]. The complex composed of GKY25 bound to CD14 was then subjected to the same solvation, minimization, and equilibration procedures as described above, after which their binding energies were determined using MMPBSA. A similar protocol was utilized for the addition of a staple at positions i and i + 4. The difference in binding energies to the non-stapled GKY25 was then calculated.

### Molecular dynamics simulation of CD14 and sHVF18

All-atom molecular dynamics (MD) simulations were performed to understand the dynamics of sHVF18 binding to CD14. The starting coordinates for sHVF18 were obtained from NMR spectroscopy and further modeled using CHARMM-GUI Solution Builder[1]. The hydrophobic staple was added by mutating residues 10 and 14 to alanine and linking them with two pentene segments. To model the binding of sHVF18 to CD14, we performed an unbiased protein-protein docking using the ClusPro web server[7]. No intermolecular restraints were applied during the docking protocol. A total of 17 docking poses were generated, with 11 poses found at the N-terminal hydrophobic pocket of CD14 (Supplementary Fig. 14a), similar to our previous study[8]. These 11 poses were then subjected to a short MD simulation to test the stability of peptide binding. Peptide-bound CD14 was solvated with TIP3P water molecules, and 0.15 M NaCl salt was added to neutralize the system. Energy minimization was performed using the steepest descent method, and a short 125 ps equilibration simulation, whereby positional restraints with Fc of 400 kJ mol$^{-1}$ nm$^{-2}$ were applied to the backbone atoms, was conducted following the standard CHARMM-GUI protocols[9]. A 200-ns MD simulation was performed for each pose using GROMACS 2021[10] and the CHARMM36m force field[67]. The temperature of the system was maintained at 310 K by coupling to the Nosé-Hoover thermostat[12,13], while the pressure was maintained at 1 atm by isotropic coupling to the Parrinello-Rahman barostat[14]. The particle mesh Ewald (PME) method was used to compute the electrostatic interactions, while the van der Waals interactions were truncated at 1.2 nm with a force switch function applied between 1.0 to 1.2 nm. All covalent bonds with hydrogen atoms were constrained using the LINCS algorithm[15], and a 2 fs integration time step was used.

We then determined whether sHVF18 remained bound to CD14 in each predicted binding pose by calculating the minimum distance between the peptide and the protein (Supplementary Fig. 14b). To determine the most probable binding pose, we concatenated all 11 simulation trajectories, and performed clustering analysis using the GROMOS method with an RMSD cut-off of 0.4 nm. The central structure of the most populated cluster (representing around 90% of the trajectories) was selected for further MD simulations. Three independent 1-μs simulations with different initial velocities were performed for this structure using the same protocol described above. As a control, three independent 1 μs simulations of CD14 without the bound sHVF18 peptide were conducted using the same protocols.

### Molecular dynamics simulation of lipid A and Ra LPS with sHVF18

A multiscale MD simulation approach was used to understand how sHVF18 interacts with LPS via its lipid A anchor. The NMR structure of sHVF18 was converted to coarse-grained (CG) representation using the MARTINI 2.2 force field with an elastic network using the standard ElNeDYn parameters applied to maintain the secondary and tertiary structures.[16] The hydrophobic staple was introduced via two additional beads linking the backbone beads of residues 10 and 14. The CG parameters were validated by comparing the average distances between the backbone beads of all residues to equivalent distances of Cα atoms from atomistic simulations of the peptide in solution (Supplementary Fig. 24). A large lipid A aggregate was generated by randomly placing 60 lipid A molecules into solution and allowing them to self-assemble during a 1 μs simulation. As divalent cations are required to cross-link the phosphate head groups of adjacent lipid A molecules, neutralizing Ca$^{2+}$ ions was included in the simulation. The Ca$^{2+}$ bound lipid A aggregate from the end of this simulation was then positioned inside a larger box, whereby 30 copies of the sHVF18 peptides were added at a minimum distance of 2 nm away from the surface of the lipid aggregate. The system was re-solvated with standard MARTINI water particles and neutralized with an additional 0.15 M NaCl salt. Energy minimization was performed using the steepest descent method. A 50 ns equilibration simulation was conducted whereby positional restraints with force constants of 1000 kJ mol$^{-1}$ nm$^{-2}$ and 500 kJ mol$^{-1}$ nm$^{-2}$ were applied to the peptide backbone and to lipid A, respectively. Three independent 10 μs production runs were then conducted. The temperature was maintained at 310 K using the velocity-rescaling thermostat[17], while the pressure

was maintained at 1 atm by isotropic coupling to a Parrinello-Rahman barostat. Electrostatic interactions were computed using the reaction field method, while the van der Waals interactions were truncated using the potential shift Verlet scheme, with a 1.1 nm short-range cut-off applied to both. A 10-fs integration time step was used for the CG simulations.

We repeated our CG simulation with *E. coli* rough Ra LPS molecules (lipid A plus 10 core sugars). We placed 60 copies of Ra LPS molecules randomly into solution and allowed it to self-assemble into a large LPS aggregate during a 1 μs simulation. Similar to our lipid A simulations, $Ca^{2+}$ ions were included in the system to intercalate between LPS headgroups. The $Ca^{2+}$-bound LPS aggregate was extracted from the end of the simulation and 30 copies of sHVF18 were added at least 2 nm from the surface of the lipid aggregate. The system underwent the same solvation, energy minimization and equilibration protocols as described above for sHVF18 simulations with lipid A. Similarly, three independent 10 μs production simulations were performed.

To validate the results from our CG simulations, we performed atomistic simulations of a similar lipid A-peptide system. A small lipid A aggregate was generated by placing 12 lipids A molecules in solution and running a 200 ns simulation to allow them to self-assemble. The coordinates and parameters for *E. coli* lipid A were obtained from the CHARMM-GUI[9]. Similar to the CG system, divalent $Ca^{2+}$ ions were added. The $Ca^{2+}$ bound lipid A aggregate was then extracted from the final snapshot of the simulation, and a single sHVF18 peptide was placed ~2 nm from the surface of the lipid aggregate. The system was re-solvated with TIP3P water molecules and neutralized with 0.15 M NaCl salt. Energy minimization, equilibration and production runs were performed using the parameters described above for atomistic simulations of the CD14-peptide complex, except that the peptide was not subject to distance and dihedral angle restraints in these simulations to avoid further bias. Three independent replicates of 1 μs production simulations were conducted. Simulations were visualized using VMD[18], and analyses of trajectories were performed using GROMACS 2021[10].

### Mouse model of subcutaneous inflammation

The immunomodulatory effects of HVF18 and sHVF18 were studied in BALB/c tg(NF-κB-RE-Luc)-Xen reporter mice (Taconic, 10–12 weeks old). The dorsum of the mice (5–8 mice per treatment group) was carefully shaved and cleaned. The peptide (50 μg mouse⁻¹) was injected subcutaneously simultaneously with *E. coli* LPS (25 μg mouse⁻¹) in a final volume of 200 μL. After injection, animals were immediately transferred to individually ventilated cages and imaged 3 h later. We used an In Vivo Imaging System (IVIS Spectrum, PerkinElmer Life Sciences) for the determination of NF-κB activation. Fifteen minutes before the IVIS imaging, mice were intraperitoneally given 100 μL of D-luciferin (150 mg/kg body weight). Bioluminescence from the mice was detected and quantified using Living Image 4.0 Software (PerkinElmer Life Sciences).

### Mouse model of systemic inflammation

*E. coli* 0111:B4 LPS was resuspended in 10 mM Tris at pH 7.4. A sublethal dose (6 mg per kg of body weight) was injected intraperitoneally (i.p.) into male C57BL/6 mice (11–12 weeks, 22 ± 5 g). Thirty minutes after, 10, 20, 50, 100 or 500 μg sHVF18 (in 10 mM NaOAc at pH 5) was administrated i.p. into the mice. After 8 and 20 h post LPS challenge, mice were deeply anesthetized by isoflurane, and the blood was collected by cardiac puncture and stored at −80 °C until further analysis. For histology and scanning electron microscopy (SEM) analyses, lungs were prepared as described below. For transcriptome analyses, the lungs were collected in RNA*later*® (Invitrogen, Carlsbad, CA, USA) and stored at −20 °C.

### Histology

Lung tissues collected from mice or pigs were fixed overnight in 10% neutral buffered formalin solution (Sigma Aldrich, St. Louis, MI, USA) and fixed at 4 °C overnight. Formalin-fixed tissues were processed with a graded ethanol series (solutions obtained from Histolab Products AB, Gothenburg, Sweden) and clearing solution (Sigma Aldrich, St. Louis, MI, USA) prior to paraffin embedding (Histolab AB, Askim, Sweden). Sections of 4-μm were cut and transferred to SuperFrost Plus microscopy slides (Thermo Fisher Scientific, Waltham, MA, USA). Sections were dried at room temperature overnight. Sections were stained with hematoxylin and eosin (H&E, Histolab AB, Askim, Sweden) after de-paraffinization, followed by dehydration in consecutively graded ethanol and xylene solutions (Histolab AB, Askim, Sweden). Stained sections were mounted with Pertex solution (Histolab AB, Askim, Sweden). Bright-field images were acquired with an Olympus CKX53 microscope (Olympus, Shinjuku, Tokyo, Japan). For histology scoring, an acute lung injury system recommended by the American Thoracic Society was used[68]. Scores were given based on a scale of 0 to 6. The scores from three independent blinded scorers are presented as the average of the sum of the characteristic scores for each sample.

### Scanning electron microscopy (SEM)

Lung tissues were fixed overnight at 4 °C in "SEM fix" (0.1 M Sorenson's phosphate buffer at pH 7.4, 2% formaldehyde and 2% glutaraldehyde). After fixation, samples were washed twice in 0.1 M Sorenson's buffer (pH 7.4). Washed samples were dehydrated in a graded series of ethanol (50%, 70%, 80%, 90%, and twice in 100%) and critical point dried. Samples were mounted on 12.5-mm aluminum stubs and sputtered with 10 nm Au/Pd (80/20) in a Quorum Q150T ES turbo-pumped sputter coater. Finally, prepared samples were examined in a Jeol JSM-7800F FEG-SEM (JEOL, Japan).

### Global gene expression

Total RNA was extracted from murine lungs in RLT buffer with 1% β-mercaptoethanol after disruption in a tissue homogenizer (Tissue-Lyser LT, Qiagen) using Ceramic Beads 1.4 mm (Qiagen) with the RNeasy Mini Kit (Qiagen). The library was prepared with 500 ng of RNA using TruSeq® Stranded mRNA Library Prep Kit (20020594, Illumina). The quality of the library was evaluated by the High Sensitivity Screen Tape Kit (5067-5579, Agilent). Sequencing (0.7 nM of the library) was performed using QuantIT 1X dsDNA HS Assay Kit (Q33232) on a NovaSeq with 2 × 100 bp pair reads to an average depth of 31 M read pairs per sample.

### Bioinformatics

Transcriptomic data were aligned to the GRCm38 genome using gencode version 25 as the gene model using STAR (2.7.6a). Quality control was performed by collating data from STAR and Picard with no sample failing quality control (see Supplementary Fig. 20a). Gene quantification was done using featureCounts[69]. For downstream analysis, R (version 4.0.3) was used, and plotting was done using ggplot2 (2.3). Pre-processing, including the transformation of count data to log2-counts per million (logCPM), was done using the voom-function (limma version 3.46). Differential expression analysis was carried out by fitting a linear model using "untreated 8 h" as a baseline. Transcription factor and GO enrichment analysis was done using the clusterProfiler package (3.18) using MSigDB collections C3 and C5. A Venn diagram was plotted using eulerr.

### Cecal-ligation puncture (CLP) model in mice

A standard and widely used Cecal-ligation and puncture (CLP) model of sepsis[70,71] was employed. All the procedures were done under aseptic conditions. Briefly, 7–8 weeks old C57BL/6 mice (Janvier-Labs,

France) were anesthetized with an i.p. injection of ketamine (60 mg kg$^{-1}$) and xylazine (10 mg kg$^{-1}$). The abdomen was shaved with a rodent clipper and disinfected with a povidone-iodine solution, followed by wiping with a 70% alcohol swab. A midline laparotomy was performed by making a 1–1.5 cm incision. Cecum was exposed by pulling out gently through the incision. The cecum was ligated with a 6.0 silk suture and perforated twice with a 22-gauge needle. The cecum was gently squeezed to extrude a small amount of fecal material from the perforated sites and then returned to the peritoneal cavity. The peritoneum was closed with a 6.0 silk suture, followed by skin closure with skin wound clips (AgnThos, Lidingö, Sweden). Mice were resuscitated by subcutaneously injecting 1 mL of prewarmed saline, kept on a heating pad for recovery, and returned to their respective cages immediately after. The mice were i.p. treated twice daily with 100 µg of sHVF18 (in 100 µL water) for a total of 5 days. The first dosage was given 30 min after the CLP procedure. The control group was given 100 µL of water i.p.

## Porcine ARDS model

Female and male adult farm-raised wild-type American Yorkshire pigs (Sus scrofa domesticus) were used. The animals were stratified into either treatment or non-treatment groups. A total of ten pigs with a mean weight of 45 kg were premedicated with ketamine (Ketaminol® vet. 100 mg mL$^{-1}$; Farmaceutici Gellini S.p.A., Aprilia, Italy; 20 mg kg$^{-1}$) and xylazine (Rompun® vet. 20 mg mL$^{-1}$; Bayer AG, Leverkusen, Germany; 2 mg kg$^{-1}$). A urinary catheter was inserted into the bladder. A peripheral intravenous (IV) line was placed in the earlobe, and general anesthesia was maintained with ketamine (Ketaminol® vet, MSD Animal Health Sweden, Stockholm, Sweden), midazolam (Midazolam Panpharma®, Panpharma Nordics AS, Oslo, Norway), and fentanyl (Leptanal®, Piramal Critical Care B.V., Voorschoten, Netherlands) infusions. A Siemens-Elema ventilator (Servo 900 C, Siemens, Solna, Sweden) was used to establish mechanical ventilation with a 7.5-size endotracheal tube for intubation. Volume-controlled ventilation (VCV) with the flow pattern switch in "constant flow" was used, which lowers the peak pressures according to the manufacturer's instructions. To achieve an I:E ratio of 1:2, inspiration time was set to 25% with a pause time of 10%, and ventilation was adjusted to maintain carbon dioxide levels (PaCO$_2$) between 33–41 mmHg. The tidal volume (Vt) was kept at 6–8 mL kg$^{-1}$. The Eq. (2) was used to determine the dynamic compliance.

$$C_{dyn} = \frac{V_T}{(\text{peak pressure} - \text{PEEP})} \qquad (2)$$

Further, an arterial line (Secalon-T™, Merit Medical Ireland Ltd, Galway, Ireland) was inserted in the right common carotid artery. A pulmonary artery catheter (Swan-Ganz CCOmbo V and Introflex, Edwards Lifesciences Services GmbH, Unterschleissheim, Germany) was placed in the right internal jugular vein.

To induce an ARDS according to the Berlin criteria[40] *E. coli* LPS (O111:B4, Sigma-Aldrich, Merck KGaA, Darmstadt, Germany) was used intravenously, as previously described[38,39]. Prior to administration, LPS was diluted in saline solution (2 µg kg$^{-1}$ min$^{-1}$). As a result of the LPS administration, all animals developed hemodynamic instability and required continuous inotropic support, provided as an infusion of norepinephrine (40 µg mL$^{-1}$, 0.05–2 µg kg$^{-1}$ min$^{-1}$; Pfizer AB, Sollentuna, Sweden) and dobutamine (2 mg mL$^{-1}$, 2.5–5 µg kg$^{-1}$ min$^{-1}$; Hameln Pharma Plus GmbH, Hameln, Germany). Ringer's acetate (Baxter Medical AB, Kista, Sweden) was used to compensate for fluid loss. The different ARDS stages were defined based on the measured PaO$_2$ FiO$_2^{-1}$ ratio, according to the Berlin definition of ARDS[40]: Mild ARDS for a ratio between 201–300 mmHg, moderate ARDS for a ratio between 101–200 mmHg, and severe ARDS for a ratio ≤100 mmHg. The ARDS state was considered as confirmed when two separate arterial blood gas measurements, taken within a 15-min interval, fell within the Berlin definition's PaO$_2$ FiO$_2^{-1}$ range.

In the treated cohort, each animal received two intravenous doses of sHVF18 solution (12 mg in 50 mL) over the course of 30 min, administrated using the central venous catheter in the superior vena cava. Three hours was the predetermined endpoint; if any animal died before this endpoint, the last observed time point was considered the endpoint for the data analysis.

## Arterial blood gas analysis in pigs

Arterial blood from pigs was collected every 30 min and analyzed with an ABL 90 FLEX blood gas analyzer (Radiometer Medical ApS, Brønshøj, Denmark). According to clinical standards, the measurements were normalized to a blood temperature of 37 °C.

## Hemodynamic measurements in pigs

The animals were closely observed, and hemodynamic parameters were measured and recorded before the start of ARDS induction and every 30 min thereafter, using thermodilution with a Swan-Ganz catheter and an arterial line. Parameters recorded were heart rate (HR), systolic blood pressure (SBP), diastolic blood pressure (DBP), mean arterial pressure (MAP), central venous pressure (CVP), cardiac output (CO), systolic pulmonary pressure (SPP), diastolic pulmonary pressure (DPP), mean pulmonary pressure (MPP), pulmonary artery wedge pressure (PAWP), systemic vascular resistance (SVR), and pulmonary vascular resistance (PVR).

## Statistical analysis

All microbiological and cell culture-based assays showed biological replicates and were repeated at least three times. Data are presented as means ± SD or SEM. Differences in the mean between the two groups were analyzed using Student's *t*-test for normally distributed data and the Mann-Whitney test otherwise. To compare means between more than two groups, a one-way ANOVA with post hoc (Tukey) for normally distributed data or the Kruskal-Wallis test with post hoc (Dunn's) was used otherwise. Statistical analysis, as indicated in each figure legend, was performed using GraphPad Prism software v8. *P* values < 0.05 were considered statistically significant.

## Reporting summary

Further information on research design is available in the Nature Portfolio Reporting Summary linked to this article.

## Data availability

The MS raw files have been deposited to the ProteomeXchange Consortium via the MassIVE partner repository (reference ID: MSV000090815, [https://massive.ucsd.edu/ProteoSAFe/dataset.jsp?task=20d9f7e7f6434abfa3d56e0bf4eedf99]). The MassIVE DOI for this deposition is 10.25345/C5KD1QQ7S. Same data are available as Supplementary Data 1. The molecular dynamics trajectories for the initial and final configurations have been deposited in the Zenodo database [https://zenodo.org/record/8267913]. The Zenodo DOI for this deposition is 10.5281/zenodo.8267913. The NMR data have been deposited to the wwPDB. The PDB code for this deposition is 8BWW. The PDB DOI for this deposition is 10.2210/pdb8bww/pdb. The BMRB code for this deposition is 34778. PDB data: 4GLP, 1WWL, 5Z5X. Source data are provided with this paper.

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

## Acknowledgements

This work was supported by grants from the Swedish Research Council (project 2017-02341, 2020-02016, 2021-06388)—A.S., Edvard Welanders Stiftelse and Finsenstiftelsen (Hudfonden)—G.P. and A.S., the Royal Physiographic Society—G.P., the Crafoord, Österlund and Mats Paulsson Foundations—A.S, Vinnova—A.S., and the Swedish Government Funds for Clinical Research (ALF)—A.S., the FY21_CF_HTPO SEE-D_ID_BII_C211418001 grant funded by A*STAR and BII (A*STAR) core funds —P.J.B. Support from the Swedish National Infrastructure for Biological Mass Spectrometry (BioMS) and the SciLifeLab, Integrated Structural Biology platform is gratefully acknowledged for support with mass spectrometry. We thank the Center for Translational Genomics (Lund University) and Clinical Genomics Lund, SciLifeLab for providing sequencing service, and the Lund University Bioimaging Centrum (LBIC) for access to electron microscopy facilities. Petter Storm is gratefully acknowledged for skillful support with bioinformatic analyses. In silico simulations were performed on resources of the National Supercomputing Centre, Singapore (https://www.nscc.sg), the A*STAR Computational Resource Centre (A*CRC) and the supercomputer Fugaku provided by RIKEN through the HPCI System Research Project (Project ID: hp220297). We acknowledge BioRender's assistance in creating cartoons.

## Author contributions

G.P. and A.S. conceptualized and designed the study. M.P. and G.P. designed and conducted the in vivo mouse experiments. G.P., A.-C.S, and J.P. performed the in vitro experiments. F.S. and P.J.B performed in silico docking and multiscale simulations. S.E. performed HDX analysis. C.D. and B.W. performed NMR analyses. S.K. performed mass spectrometry analyses. F.O., M.M., S.H., D. E., and S.L. performed the porcine in vivo studies. G.P. and A.S. wrote the manuscript. All co-authors contributed respective text parts and data analyses and revised the manuscript.

## Funding

## Competing interests

The authors declare the following competing interests: A.S. is a founder of in2cure AB, a parent company of Xinnate AB, companies that are developing therapies based on thrombin-derived peptides and variants. G.P. is employed part-time (20%) by Xinnate AB. The TCP family, including stapled variants, is patent protected. Patent applicant: In2cure AB, Sweden. Inventors: Ganna Petruk, Artur Schmidtchen. Status of application: Publication Number: WO/2023/067167, Publication Date: 27.04.2023, International Application No.: PCT/EP2022/079429, International Filing Date: 21.10.2022. Specific aspect of manuscript covered in patent application: The invention relates to thrombin derived peptides comprising at least one internal covalent linkage between the side chains of two non-neighboring, internal amino acids. The peptides have anti-inflammatory effect. The sequences described herein are included in the patent application. The other authors declare no financial or non-financial conflicts of interest.
