## [Peer Review File · Nature Communications]

REVIEWER COMMENTS

Reviewer #1 (Remarks to the Author):

The manuscript by Petruk et al describes in great detail the discovery of a peptide with the potential to treat systemic bacterial sepsis. The authors have been pioneers in identifying the role of thrombin fragments in modulating post injury inflammatory processes. In this study the authors begin with a C-terminal peptide GKY25 currently in a Phase 1 clinical trial to evaluate its safety in suction blisters, with its ultimate intended use as a topical therapy. To improve the stability of this peptide against proteases, the authors modify the molecule using covalent stapling. Following optimization, they arrive at a truncated stapled peptide, sHVF18, which is the primary subject of the manuscript. Applying an extensive array of biophysical techniques, Petruk et al demonstrate clearly that sHVF18 interacts with both LPS and CD14 with reasonable affinity and could exhibit desirable therapeutic characteristics. The authors demonstrate in several animal models, all well established models of bacterial sepsis, that sHVF18 administration appears to be beneficial in these preclinical settings.

I think the manuscript is well written and the methods well described.

I do believe, however, that one of the reasons we continue to fail in our attempts to treat sepsis, is that the models used do not represent sepsis as it occurs in humans. In the animal models used here, and in most preclinical studies, both the LPS challenge and the dose of therapeutic are carefully titrated, in the sense that just enough LPS is administered to achieve an effect, but not too much. A slight reduction in the administered dose of LPS generally has a very minimal effect. In addition, in almost every case, the LPS administration (or bacterial insult) is timed to occur either simultaneously with the administration of the therapeutic, or soon after. As a result whatever benefit we might observe could be due to the prevention of the onset of sepsis (by binding up LPS, for example in this case). Unfortunately, in almost all clinical settings, sepsis has already begun. Any therapeutic of value must be able to put a brake on the ongoing process.

This is not a criticism of the submitted manuscript, but my own concerns regarding the development of drugs to treat sepsis.

Reviewer #2 (Remarks to the Author):

Sepsis is a systemic inflammation caused by bacteria and is among the most common causes of death in intensive care units. The etiology of sepsis is quite complicated as it involves the activation of several inflammatory pathways that comprise multiple pathogen-associated molecular patterns. As a result, targeting bacteria using antibiotics solves the infection problem only partially and does not address

inflammation. In past years, a few drugs have been utilized in clinical settings but have withdrawn from the market due to low efficacy. In this manuscript, Schmidchen and co-workers report a new stapled polypeptide active toward TLR-driven systemic inflammation. The synthetic peptide was inspired by the natural sequence from the C-terminus of thrombin (GKY25). The GKY25 sequence is reminiscent of many antimicrobial peptides with an alternation of hydrophobic and hydrophilic residues. As with all antimicrobial peptides, GKY25 can be degraded *in vivo*. Therefore, the authors utilized a technique developed by Kamysz in 2018 on antimicrobial peptides to 'staple' the side chains of the peptide and delay its degradation. This stapled peptide was then further optimized to obtain a protease-robust sequence to be tested *in vivo* on animal models. The first part of the manuscript deals with the biophysical characterization of the stapled peptide using spectroscopic and calculation techniques (CD, NMR, molecular docking, MD simulations, etc.) as well as biochemical assays to establish its binding affinity for its target CD14. The second part deals with the *in vivo* assays, showing the peptide's biological activity and efficacy.

Overall, this study can be suitable for the broad audience of Nature Communication as it reports a promising approach to counteract sepsis. However, I am concerned about the biophysical characterization of the stapled peptide free in solution and bound to its target CD14. To obtain the final model of interactions with CD14, the authors characterized the free peptide in TFE (trifluoroethanol)/water mixture using NMR spectroscopy. Then they utilize the conformers generated by NMR-derived restraints to model the molecular complex and utilize its coordinates for MD simulations. While the atomistic and coarse-grained simulations are carried out with state-of-the-art techniques, the starting configuration of these simulations is concerning. As I mentioned, the authors start from models derived from NMR structures of the free state, which might differ significantly from the bound conformation. Additionally, the NMR studies were carried out in an organic solvent mixture. As stated by the authors, TFE induces helical structures in linear or constrained peptides and might result in a suboptimal conformational ensemble. Also, it is unclear how the docking studies were performed. Specifically, the authors do not describe the intermolecular restraints utilized to drive the molecular docking protocol. If only the protection factors from the HD/MS were used, the molecular model would be underdetermined. The initial configuration of the complex is critical to structural and dynamic characterization of the complex as the authors utilize it to carry out MD simulations and describe the motions of the complex.

Other concerns:

- A) The authors should report the statistics on the structural calculations of the peptides (# restraints, i.e., # of ROEs, # of dihedral angles, etc.
- B) The data from HD/MS data are difficult to read and interpret from the supporting information figures.
- C) There are a few NMR studies on the innate immune receptor CD14. The amide fingerprint of the protein is well resolved (see BBRC 368, pages 231-237, 2008). Studying the peptide/CD14 interactions by NMR spectroscopy might be doable and would overcome the uncertainty of the initial model built only on a few intermolecular restraints.

Reviewer #3 (Remarks to the Author):

The authors report a novel peptide design called sHVF18, which binds to CD14 and demonstrates its therapeutic potentials through in vivo experiments. The binding interaction is comprehensively characterized through multiple approaches including NMR, CD, HDX-MS, and simulations. Regarding on HDX-MS, here are some comments for the authors to consider.

1. Lines 169-176 as well as Figure 2d illustrates the conformational differences between linear and stapled HVP18. I would recommend the authors to report deuterium uptake together with the MS spectra for a better visualization.
2. In Figure 2d, there seems to be some mismatch between the two spectra in the bottom row. The isotopic distribution between the left and the right panels does not seem to match, with more signals from the right-hand side panel (stapled HVP18). This can be a result of elevated deuterium uptake, and it will be helpful for the authors to show the fitted/theoretical isotopic distribution generated by HDEaminer.
3. In Supplementary Figures 11 to 13, authors present the differential HDX plots and the butterfly plots, highlighting the differences in terms of deuterium uptakes between bound and unbound states. In each of the butterfly plots, authors highlight a $\pm 0.5D$ interval, which seems to suggest any value beyond this interval is considered significant. Authors should comment on how this $0.5D$ value is obtained. Moreover, authors should also perform statistical analysis of the HDX data as suggested in Ref. 33, and determine if the difference in deuterium uptake is significant or not from a statistical point of view. This should take into consideration the propagated error of the two replicates for each uptake time under both bound and unbound states, and it is essential to make statistical justifications before reaching any HDX-based conclusions.
4. Lines 217-272, authors discussed the LPS-sHVF18 interaction based on an early converging of sHVF18 - CD14 binding kinetics. This observation can also be explained as the sHVF18 - CD14 binding kinetics is accelerated with the presence of LPS, leading to a higher sHVF18 on & off rate when binding with CD14, therefore inducing an early converging of bound vs unbound HDX kinetic curves. Such reasoning should be discussed in the manuscript/Sl. Also, if LPS-sHVF18 interaction is proposed, a HDX experiment between LPS and sHVF18 needs to be considered.
5. Line 728, authors mentioned that they were using deglycosylated CD14, has there been any literature report suggesting that glycans will not impact the CD14-peptide binding? Will deglycosylation change the binding regime/affinity between CD14 and peptides?
6. Line 734, authors mentioned "Unbound/Apo state CD14 samples constituted 2 μ L and 2 μ L MQ". Does the MQ stand for milli-Q water?

The reviewers' concerns are addressed point-by-point below:

Reviewer #1

Comment 1. The manuscript by Petruk et al describes in great detail the discovery of a peptide with the potential to treat systemic bacterial sepsis. The authors have been pioneers in identifying the role of thrombin fragments in modulating post injury inflammatory processes. In this study the authors begin with a C-terminal peptide GKY25 currently in a Phase 1 clinical trial to evaluate its safety in suction blisters, with its ultimate intended use as a topical therapy. To improve the stability of this peptide against proteases, the authors modify the molecule using covalent stapling. Following optimization, they arrive at a truncated stapled peptide, sHVF18, which is the primary subject of the manuscript. Applying an extensive array of biophysical techniques, Petruk et al demonstrate clearly that sHVF18 interacts with both LPS and CD14 with reasonable affinity and could exhibit desirable therapeutic characteristics. The authors demonstrate in several animal models, all well established models of bacterial sepsis, that sHVF18 administration appears to be beneficial in these preclinical settings.

I think the manuscript is well written and the methods well described.

I do believe, however, that one of the reasons we continue to fail in our attempts to treat sepsis, is that the models used do not represent sepsis as it occurs in humans. In the animal models used here, and in most preclinical studies, both the LPS challenge and the dose of therapeutic are carefully titrated, in the sense that just enough LPS is administered to achieve an effect, but not too much. A slight reduction in the administered dose of LPS generally has a very minimal effect. In addition, in almost every case, the LPS administration (or bacterial insult) is timed to occur either simultaneously with the administration of the therapeutic, or soon after. As a result whatever benefit we might observe could be due to the prevention of the onset of sepsis (by binding up LPS, for example in this case). Unfortunately, in almost all clinical settings, sepsis has already begun. Any therapeutic of value must be able to put a brake on the ongoing process.

This is not a criticism of the submitted manuscript, but my own concerns regarding the development of drugs to treat sepsis.

Response 1: We greatly appreciate the positive evaluation of our work by reviewer 1. We fully acknowledge and share the reviewer's concerns regarding the challenges in developing novel drugs for systemic infections and sepsis. To address these concerns and improve the presentation of our research, we have made several changes to the Introduction and Discussion sections of the manuscript.

Firstly, we provide a clearer understanding of the therapeutic effects of sHVF18 in relation to the particular models used. Secondly, we have placed increased focus on bacteria-induced

systemic inflammation, clearly defining the therapeutic efficacy of sHVF18 within specific models such as LPS-induced systemic inflammation, ARDS, or CLP-induced polymicrobial infection.

Furthermore, in response to the reviewer's important comment on challenges with translation, we have added a thorough discussion of the limitations of our study in a separate section at the end of the Discussion (*page 17, line 525-539*). This enables us to address the concerns regarding analyses in preclinical models and emphasizes the need for evaluating therapeutics that can intervene in the ongoing process of systemic infection and sepsis in clinical settings.

Once again, we sincerely appreciate reviewer 1's feedback and constructive comments.

Reviewer #2

Comment 1. Sepsis is a systemic inflammation caused by bacteria and is among the most common causes of death in intensive care units. The etiology of sepsis is quite complicated as it involves the activation of several inflammatory pathways that comprise multiple pathogen-associated molecular patterns. As a result, targeting bacteria using antibiotics solves the infection problem only partially and does not address inflammation. In past years, a few drugs have been utilized in clinical settings but have withdrawn from the market due to low efficacy. In this manuscript, Schmidchen and co-workers report a new stapled polypeptide active toward TLR-driven systemic inflammation. The synthetic peptide was inspired by the natural sequence from the C-terminus of thrombin (GKY25). The GKY25 sequence is reminiscent of many antimicrobial peptides with an alternation of hydrophobic and hydrophilic residues. As with all antimicrobial peptides, GKY25 can be degraded *in vivo*. Therefore, the authors utilized a technique developed by Kamysz in 2018 on antimicrobial peptides to 'staple' the side chains of the peptide and delay its degradation. This stapled peptide was then further optimized to obtain a protease-robust sequence to be tested *in vivo* on animal models. The first part of the manuscript deals with the biophysical characterization of the stapled peptide using spectroscopic and calculation techniques (CD, NMR, molecular docking, MD simulations, etc.) as well as biochemical assays to establish its binding affinity for its target CD14. The second part deals with the *in vivo* assays, showing the peptide's biological activity and efficacy.

Overall, this study can be suitable for the broad audience of Nature Communication as it reports a promising approach to counteract sepsis.

Response 1: We thank the reviewer for the positive comment, which motivated us to revise and further strengthen our manuscript.

Comment 2. However, I am concerned about the biophysical characterization of the stapled peptide free in solution and bound to its target CD14. To obtain the final model of interactions with CD14, the authors characterized the free peptide in TFE (trifluoroethanol)/water mixture using NMR spectroscopy. Then they utilize the conformers generated by NMR-derived restraints to model the molecular complex and utilize its coordinates for MD simulations. While the atomistic and coarse-grained simulations are carried out with state-of-the-art techniques, the starting configuration of these simulations is concerning. As I mentioned, the authors start from models derived from NMR structures of the free state, which might differ significantly from the bound conformation. Additionally, the NMR studies were carried out in an organic solvent mixture. As stated by the authors,

TFE induces helical structures in linear or constrained peptides and might result in a suboptimal conformational ensemble.

Response 2: The reviewer raised an important concern on peptide structural determination in organic solvent. However, based on structural comparison to our MD simulations of the peptide with lipid A aggregate, we are confident that the structure represents a physiologically relevant LPS-bound state of the peptide. We performed additional structural analyses to show that the peptide retained the structural features found in the NMR ensemble when it is bound to a lipid A aggregate throughout our simulations. These analyses have been added to Supplementary Fig. 18a-c and explained in the main text (*page 10, line 293-311*).

With regards to the concern on the final model of interactions with CD14, the reviewer is correct in saying that the structure of the free state of the peptide might differ from the CD14-bound state. To rigorously address this, we conducted completely unbiased docking (i.e. without any intermolecular restraints), and subsequently performed a series of extended MD simulations for each of 11 different poses at the N-terminal binding site, without imposing any distance/dihedral restraints derived from the NMR spectroscopy. Hence, the peptide was allowed to sample extensive conformational space in the absence of any bias from the NMR data, regardless of its starting structure. Encouragingly, our MD simulations showed a stable and consistent binding mode and, most importantly, the final representative model was in good agreement with our HDX data vis-à-vis the residues involved in peptide-CD14 interactions.

It is also noteworthy that the sHVF18 residues forming the evolutionarily conserved TCP innate motif KKWIQK were observed to be crucial to CD14 (and LPS) binding in our simulations. This emphasizes the importance of the increased stability introduced to this particular region by stapling, in agreement with our HDX and MST data.

These additions are reflected in Materials and Method under “Molecular dynamics simulation of CD14 and sHVF18” (*page 28-29, line 884, 918*) and the updated results are shown in Fig. 3d and e, as well as Supplementary Fig. 14 with text changes in the Results section (*page 8-10, line 242-292*).

Determining the structure of CD14-bound to the peptide by means of, for example, NMR or X-ray crystallography, is extremely challenging and is beyond the scope of the current study. While we agree that our data might not substitute structural determination of peptide bound to CD14, it serves as supporting evidence that the peptide binds to the expected LPS-binding site on CD14, similar to the native non-stapled version, and hence explains the anti-inflammatory effects observed in vitro and in vivo.

Comment 3. Also, it is unclear how the docking studies were performed. Specifically, the authors do not describe the intermolecular restraints utilized to drive the molecular docking protocol. If only the protection factors from the HD/MS were used, the molecular model would be underdetermined. The initial configuration of the complex is critical to structural and dynamic characterization of the complex as the authors utilize it to carry out MD simulations and describe the motions of the complex.

Response 3: We agree with the reviewer that using HDX protection factors to choose our docked model might result in the model being underdetermined. Thus, we performed a completely unbiased peptide-protein docking without any intermolecular restraints. Then, we

simulated all 11 docking poses of the peptide on the N-terminal binding site of CD14, where LPS is expected to bind. A representative structure from these simulations was obtained using clustering analysis and subsequently subjected to longer microsecond timescale simulations to ensure extensive conformational sampling. Encouragingly, the results from these independently-performed docking calculations and MD simulations were in good agreement with our HDX data, and are unbiased by starting structure. We have updated the Materials and Methods section to provide more details on this process (page 28-29, line 884918).

Comment 4. The authors should report the statistics on the structural calculations of the peptides (# restraints, i.e., # of ROEs, # of dihedral angles, etc).

Response 4: The summary of conformationally restricting experimental constraints for NMR structure is given in Supplementary Table 6.

Comment 5. The data from HD/MS data are difficult to read and interpret from the supporting information figures.

Response 5: We apologise and thank the reviewer for the critical comment about data representation. We now include the excel file called Supplementary Data 1 with the main supporting information for the HDX-MS.

Comment 6: There are a few NMR studies on the innate immune receptor CD14. The amide fingerprint of the protein is well resolved (see BBRC 368, pages 231-237, 2008). Studying the peptide/CD14 interactions by NMR spectroscopy might be doable and would overcome the uncertainty of the initial model built only on a few intermolecular restraints.

Response 6: Undoubtedly NMR spectroscopy could provide more information on the peptide/CD14 interactions if CD14 could be expressed and assigned using 3D NMR experiments. In BBRC 368, pages 231-237, 2008, only the N-terminal fragment consisting of amino acids 20-172 was expressed using ¹⁵N-isotope labelled material. Since the protein was not labelled with ¹³C, no attempts of backbone assignment were made using the standard 3D NMR experiments and therefore there are no peak assignments available in the BMRB data base. For the work performed in this article, full-length CD14 isotope-labelled with ¹⁵N and ¹³C would be required for backbone assignment and given that full-length CD14 is ~300 amino acids, deuteration would also be required. Expression and isotope-labelling using yeast and purification of full-length CD14 would be a major achievement all by itself, especially given there is only one structure of human CD14 in the PDB database.

Reviewer #3

Comment 1. The authors report a novel peptide design called sHVF18, which binds to CD14 and demonstrates its therapeutic potentials through in vivo experiments. The binding interaction is comprehensively characterized through multiple approaches including NMR, CD, HDX-MS, and simulations. Regarding on HDX-MS, here are some comments for the authors to consider.

Response 1: We would like to express our gratitude to reviewer 3 for very constructive comments regarding the HDX-MS part and addressed these comments below.

Comment 2. Lines 169-176 as well as Figure 2d illustrates the conformational differences between linear and stapled HVP18. I would recommend the authors to report deuterium uptake together with the MS spectra for a better visualization.

Response 2: We fully agree with this recommendation and have added the deuterium uptake to all the MS spectra in Figure 2d and added a Supplementary Data 1. For clarity we have also changed the text (*page 6, row 172-179*) to point out this visual elements in the picture.

Comment 3. In Figure 2d, there seems to be some mismatch between the two spectra in the bottom row. The isotopic distribution between the left and the right panels does not seem to match, with more signals from the right-hand side panel (stapled HVP18). This can be a result of elevated deuterium uptake, and it will be helpful for the authors to show the fitted/theoretical isotopic distribution generated by HDEaminer.

Response 3: We agree with the reviewer, compared to the instant full uptake, as seen for HVP18, the dynamic increased uptake for sHVP18 does have a more complex isotopic pattern. The theoretical spectra is actually in the picture (a blue trace barely observable due to matching the red trace, which represents the observed distribution). Since the isotope distributions matches quite well we believe that it might be best to only show the observed isotope distribution (red) in fig 2d and then show both the theoretical and observable isotope distribution in the Supplementary Data 1, peptide tab. This ensures readability of the figure while also making the details available for the HDX experts. We hope that this constitutes a satisfactory solution.

Comment 4. In Supplementary Figures 11 to 13, authors present the differential HDX plots and the butterfly plots, highlighting the differences in terms of deuterium uptakes between bound and unbound states. In each of the butterfly plots, authors highlight a $\pm 0.5D$ interval, which seems to suggest any value beyond this interval is considered significant. Authors should comment on how this $0.5D$ value is obtained.

Response 4: Our deepest apologies, in an unfortunate effort of space saving the Supplementary Data 1 was only included in the raw data repository. We now include the Supplementary Data 1 with the main supporting information. The file contains all the information related to the HDX-MS analysis e.g. replicates and significance, as well as data for every individual peptide and state, as recommended by the HDX community in Ref. 33.

Comment 5. Moreover, authors should also perform statistical analysis of the HDX data as suggested in Ref. 33, and determine if the difference in deuterium uptake is significant or not from a statistical point of view. This should take into consideration the propagated error of the two replicates for each uptake time under both bound and unbound states, and it is essential to make statistical justifications before reaching any HDX-based conclusions.

Response 5: We fully agree with this recommendation, and we would like to highlight that the statistical analysis is a fundamental part of HDEaminer analysis. The SW automatically calculates a significance level based on replicate variance across the entire project (i.e. for both unbound and bound in a comparison). Here we used the volcano plot generated significance setting, in HDEaminer, that takes into account both replicate variance and the P-value for the individual peptide measurement. The measurements that pass both the

significance tests are then used for coloration of the heatmap, and the significance line on the butterfly plots. The reviewer's concern raises a valid point regarding the disclosure of different significance test in analysis software, we have therefore included a description of the method as a picture (also shown below) in the Supplementary Data 1, for the convenience of the readers.

In this dataset the significance was calculated to span values between 0.44-0.5 Da for the different states, but in the figure text this is simplified to 0.5 Da.

A clarifying text have been added to the figure legend of the butterfly plots in question, referring to Supplementary Data 1.

We have also made a small clarification to the experimental section (page 26, line 822-826):

In this view, each point represents a single peptide at a single time point. Mousing over a point will show you information about that point.

The x-axis of the volcano plot shows the change in the measured number of deuterons between the two selected protein states (Mutant and WT in the above image). The red vertical lines on either side of 0 indicate a significance level calculated by taking into account the variance in the replicate experiments provided. Measurements in between these two vertical red lines are not significant, because the measured difference between the two protein states does not exceed the replicate variance.

The y-axis of the volcano plot is the negative log of the p-value for that measurement (so a point high on the y-axis has a very low – and therefore significant – p-value). The horizontal red line at $y=2$ represents a 99% confidence interval (since 99% confidence requires $p \leq 0.01$, or 10^{-2}). Measurements below the horizontal red line are not significant, because their p-value is too high.

Measurements in the upper-left and upper-right areas of the volcano plot are the ones that pass both significance tests.

Comment 5. Lines 217-272, authors discussed the LPS-sHVF18 interaction based on an early converging of sHVF18 - CD14 binding kinetics. This observation can also be explained

as the sHVF18 - CD14 binding kinetics is accelerated with the presence of LPS, leading to a higher sHVF18 on & off rate when binding with CD14, therefore inducing an early converging of bound vs unbound HDX kinetic curves. Such reasoning should be discussed in the manuscript/SI. Also, if LPS-sHVF18 interaction is proposed, a HDX experiment between LPS and sHVF18 needs to be considered.

Response 5: We greatly appreciate this comment, as it prompted us to address the concern by conducting a new set of experiments to demonstrate the binding between sHVF18 and LPS. Specifically, to show LPS-sHVF18 interaction, we performed additional CG simulations of LPS aggregate with sHVF18. Indeed, the peptide adsorbed spontaneously onto the surface of an LPS aggregate with a very similar interaction profile compared to our earlier sHVF18-lipid A simulations, highlighting the importance of the lipid A component in this interaction. See Supplementary Fig. 16 and Supplementary Note 4 for details.

Furthermore, we performed microscale thermophoresis (MST) binding assay and show that sHVF18 binds to LPS with a K_d of $2.5 \pm 0.7 \mu\text{M}$, while its linear variant binds with a K_d of $4.7 \pm 0.8 \mu\text{M}$ (Supplementary Fig. 17). These new findings provide additional support for the binding between LPS and sHVF18.

The new results, along with the reviewer's observation regarding the accelerated binding kinetics of sHVF18-CD14 in the presence of LPS, have now been integrated into the main text and thoroughly discussed (page 8-10, line 242-292).

Comment 6. Line 728, authors mentioned that they were using deglycosylated CD14, has there been any literature report suggesting that glycans will not impact the CD14-peptide binding? Will deglycosylation change the binding regime/affinity between CD14 and peptides?

Response 6: The reviewer raised an interesting point. We built a structural model of CD14 with N-glycans (Figure 1) and found that both sites (N151 and N282) are distant from the LPS and peptide binding site on the N-terminus of CD14. We performed two repeats of 200 ns all-atom MD simulation of the glycosylated CD14 and indeed did not observe any interaction of the glycans with the peptide/LPS binding site. Therefore, it is unlikely for the glycans to directly impact CD14-peptide binding.

Figure 1: Glycans on CD14. CD14 is predicted to have two N-linked glycosylation sites on residues N151 and N282 (<https://glyconnect.expasy.org/browser/proteins/774>). The figure shows paucimannose glycans (green) added to the structure of human CD14 (PDB: 4GLP)

(cyan) using CHARMM-GUI. The N-terminal LPS and peptide binding site is highlighted by the dotted red circle.

Comment 7. Line 734, authors mentioned “Unbound/Apo state CD14 samples constituted 2 μ L and 2 μ L MQ”. Does the MQ stand for milli-Q water?

Response 7: Yes, it is milli-Q water. It was used to keep dilution and composition of the HDX solutions the same throughout the experiments, as the peptides were dissolved in milli-Q water. For clarity, we changed the text accordingly (*page 25, line 779*).

REVIEWERS' COMMENTS

Reviewer #2 (Remarks to the Author):

The authors addressed my concerns.